# Complex spatiotemporal oscillations emerge from transverse instabilities in large-scale brain networks

Pau Clusella [1]*, Gustavo Deco[2,3], Morten L. Kringelbach[4,5], Giulio Ruffini[6], Jordi Garcia-Ojalvo[1]

**1** Department of Medicine and Life Sciences, Universitat Pompeu Fabra, Barcelona, Spain, **2** Computational Neuroscience Group, Center for Brain and Cognition, Department of Information and Communication Technologies, Universitat Pompeu Fabra, Barcelona, Spain, **3** Institució Catalana de Recerca i Estudis Avançats (ICREA), Barcelona, Spain, **4** Department of Psychiatry, University of Oxford, Oxford, United Kingdom, **5** Center for Music in the Brain, Department of Clinical Medicine, Aarhus University, Aarhus, Denmark, **6** Brain Modeling Department, Neuroelectrics, Barcelona, Spain

* pau.clusella@upf.edu

**Data Availability Statement:** All code and data in support of this publication are publicly available at https://github.com/pclus/transverse-instabilities.

## Abstract

Spatiotemporal oscillations underlie all cognitive brain functions. Large-scale brain models, constrained by neuroimaging data, aim to trace the principles underlying such macroscopic neural activity from the intricate and multi-scale structure of the brain. Despite substantial progress in the field, many aspects about the mechanisms behind the onset of spatiotemporal neural dynamics are still unknown. In this work we establish a simple framework for the emergence of complex brain dynamics, including high-dimensional chaos and travelling waves. The model consists of a complex network of 90 brain regions, whose structural connectivity is obtained from tractography data. The activity of each brain area is governed by a Jansen neural mass model and we normalize the total input received by each node so it amounts the same across all brain areas. This assumption allows for the existence of an homogeneous invariant manifold, i.e., a set of different stationary and oscillatory states in which all nodes behave identically. Stability analysis of these homogeneous solutions unveils a transverse instability of the synchronized state, which gives rise to different types of spatiotemporal dynamics, such as chaotic alpha activity. Additionally, we illustrate the ubiquity of this route towards complex spatiotemporal activity in a network of next generation neural mass models. Altogehter, our results unveil the bifurcation landscape that underlies the emergence of function from structure in the brain.

## Author summary

Monitoring brain activity with techniques such as electroencephalogram (EEG) and functional magnetic resonance imaging (fMRI) has revealed that normal brain function is characterized by complex spatiotemporal dynamics. This behavior is well captured by large-scale brain models that incorporate structural connectivity data obtained with MRI-based tractography methods. Nonetheless, it is not yet clear how these complex dynamics

**Funding:** PC, GD, GR, and JGO have received funding from the Future and Emerging Technologies Programme (FET) of the European Union's Horizon 2020 research and innovation programme (project NEUROTWIN, grant agreement No 101017716). JGO also acknowledges financial support from the Spanish Ministry of Science and Innovation and FEDER (grant PID2021-127311NB-I00), by the "Maria de Maeztu" Programme for Units of Excellence in R&D (grant CEX2018-000792-M), and by the Generalitat de Catalunya (ICREA Academia programme). The funders had no role in study design, data collection and analysis, decision to publish, or preparation of the manuscript.

**Competing interests:** I have read the journal's policy and the authors of this manuscript have the following competing interests: GR is a co-founder of Neuroelectrics, a Company that manufactures tES and EEG technology. The remaining authors don't have any conflict of interest.

emerge from the interplay of the different brain regions. In this paper we show that complex spatiotemporal dynamics, including travelling waves and high-dimensional chaos can arise in simple large-scale brain models through the destabilization of a synchronized oscillatory state. Such transverse instabilities are akin to those observed in chemical reactions and turbulence, and allow for a semi-analytical treatment that uncovers the overall dynamical landscape of the system. Overall, our work establishes and characterizes a general route towards spatiotemporal oscillations in large-scale brain models.

## Introduction

The interplay between spiking neurons across the brain produces collective rhythmic behavior at multiple frequencies and spatial resolutions [1, 2]. This oscillatory neural activity is fundamental for proper cognitive function [3, 4], and is reflected in a plethora of spatiotemporal phenomena in recorded signals [5–8]. At the microscopic scale, computational and mathematical models have successfully captured some of these emergent properties by analyzing the collective behavior of networks of coupled neurons [1, 9–11]. In larger scales, the task becomes increasingly challenging, as one needs to model several populations of neurons, which increases the mathematical complexity and the computational cost of the problem. Mesoscale neural-mass models (NMMs) allow to overcome this situation by capturing the neural activity of large numbers of neurons with a few equations [12–17].

Recent advances on neuroimaging allow to characterize the structural organization of the brain as a complex network [18–20]. In this representation, each node of the network corresponds to a brain region composed of densely inter-connected neurons, and edges across nodes represent pairwise interactions across distant regions. Combining NMMs with connectomics data one can create large-scale brain models whose dynamical properties reflect the principles underlying macroscopic neural activity [21–24]. This framework has been used, for instance, to unveil the nature of resting-state fluctuations [25–30], investigate the relation between structural and functional connectivity [31–35], and characterize transitions between brain states [36–39]. In clinical applications, large-scale brain models allow for studying macroscopic aspects of neurophatologies such as epilepsy or Alzheimer disease [40–43], and simulate the use non-invasive brain stimulation protocols for potential treatments [44, 45].

Despite all this progress, many aspects about the mechanisms through which large-scale brain models reproduce macroscopic neural dynamics are still unknown. For instance, synaptic delays between distant regions [25, 46], noise [25, 29], and heterogeneities [28] are usually acknowledged as a source of dynamical complexity. Nonetheless, spatiotemporal behavior can also arise from homogeneous deterministic systems with instantaneous interactions [30, 35, 44]. In particular, Forrester et. al. [35] recently investigated the dynamics of a large-scale brain model composed of weakly-coupled Jansen's NMMs. By means of a phase-reduction approximation, they unveil phase-locked states emerging from an instability of the synchronized state (see also [33]). For arbitrary coupling values, however, bifurcation studies of whole-brain networks are rather limited to numerical investigations [30, 44]. Other studies show that systems of NMMs coupled through simplified network topologies might display travelling waves and even chaotic dynamics [47–50]. Whether these results translate to irregular brain networks remains, so far, unexplored.

In this paper we characterize the onset of spatiotemporal dynamics in a simple large-scale brain model without heterogeneities, noise, nor delays. In close analogy to pattern-formation mechanisms in reaction-diffusion systems [51–55], coupling among brain regions alone is

enough to spontaneously destabilize an underlying synchronized state, thereby generating complex oscillatory behavior. Our analysis consists of two parts: First we show that, given a common normalization on the incoming input of each region [56, 57], the network possesses an invariant homogeneous manifold, i.e., a set of states in which the behavior of each node is identical across all the network. These states are described by a low-dimensional system: a self-coupled version of the NMM used for the evolution of each brain region. Bifurcation analysis of the system reveals how different system parameters modify the onset of synchronized oscillatory states within the manifold.

Second, we employ the Master Stability Function (MSF) formalism [58–62] to investigate the stability of the homogeneous states to heterogeneous perturbations. The synchronized oscillatory solution of the system turns out to be transversally unstable in a large region of parameter space, giving rise to complex spatiotemporal dynamics including travelling waves, multistability, and high-dimensional chaos.

In order to illustrate the ubiquity of this mechanism, we mainly use the Jansen NMM for the single-node dynamics [15, 16, 63], but also briefly review the onset of chaotic dynamics using a next generation NMM [17]. Moreover, we also show that, even when the normalization condition is not fulfilled, the bifurcation diagram of the system remains similar to that of the simplified model. Our work extends previous findings on weakly-coupled NMMs [33, 35] and simplified network topologies [47–50] to a comprehensive characterization of the emerging spatiotemporal behavior.

## Results

### The large-scale brain model

We build a large-scale brain model using a structural connectivity (SC) network comprised of $N = 90$ brain regions defined by the Automated Anatomical Labeling parcellation (AAL-90) [64], which includes 76 cortical and 14 subcortical regions. Each brain region is represented by a network node, and pairwise interactions between regions are given by a row-normalized connectivity matrix $\tilde{W} = (\tilde{w}_{ij})$ where $i, j = 1, \ldots, N$. The connection weights $\tilde{w}_{ij}$ are non-negative quantities that indicate the synaptic strength (average number of connections) from region $j$ to region $i$, and have been obtained from tractography data from 16 human subjects (see [65] and also Methods). Although networks obtained from DTI are usually symmetric prior row-normalization, our approach can also be applied to asymmetric networks (see Methods).

In order to model the dynamics of each brain region we use, for most of the paper, Jansen's model for a cortical column [15, 16, 63]. According to this model, the behavior of each brain region is given by a system of six ordinary differential equations (Eq (9) in Methods) that account for the interactions between a population of excitatory pyramidal neurons (PNs), a population of inhibitory interneurons (INs), and recurrent connections within pyramidal neurons (rPNs). Within each region, the PNs receive two sources of external input: a baseline firing rate $p$, which we assume constant and identical across the network, and the incoming firing rates from the PNs of other brain regions, modulated by a coupling parameter $\epsilon$. Hence, long range connections established by the structural connectivity matrix are all assumed excitatory, whereas inhibition acts only locally within each network node.

Overall, we obtain a large-scale brain model composed of $90 \times 6 = 540$ ODEs (see Eq (10) in Methods). Despite its high dimensionality, this system is relatively simple to analyze, as it does not include noise nor time delays and its parameters are assumed to be identical across brain regions. Similar large-scale brain models based on the Jansen system have been analyzed in previous works, the main difference being the connectivity data used for the underlying

network topology [35, 44]. Our work extends the results of these previous studies, providing a comprehensive bifurcation landscape of the network.

## Bifurcation diagram

Following a customary approach in the literature [30, 33, 35, 44, 45, 56, 57], our definition of the structural connectome matrix $\tilde{W}$ establishes that all network nodes receive the same amount of input, i.e., $\sum_{j=1}^{N} \tilde{w}_{ij} = 1$ for all $i = 1, \ldots, N$. This condition ensures that the system has *homogeneous* (uniform) states, i.e., states in which all network nodes evolve identically. If the system is initialized in such a state, it will remain there forever unless perturbed: mathematically speaking, these states lie on an *invariant manifold*. The existence of homogeneous solutions does not prevent the existence of *heterogeneous* states, in which the trajectories of different nodes evolve differently. If a homogeneous state proves to be unstable to arbitrary (heterogeneous) perturbations, complex spatiotemporal dynamics might arise.

In this section we investigate the homogeneous states of the large-scale brain model systematically, as a means to unveil the emergence of non-trivial heterogeneous patterns. This analysis requires two steps: First, we identify the homogeneous states of the system and their stability to uniform perturbations, i.e. we study the homogeneous invariant manifold of the system. Then, we analyze the stability of these states to arbitrary perturbations using the Master Stability Function (MSF) formalism [58].

**Dynamics of the homogeneous invariant manifold.** If all brain regions behave identically, then each network node follows the dynamics of the self-coupled Jansen model given by Eq (18) in Methods. This system defines the homogeneous invariant manifold of the system, hence its stable states correspond to those homogeneous states of the full model (Eq 10) that are stable to uniform perturbations. In this section we analyze the dynamics of Eq (18) alone. Therefore, the term *stability* in this section refers to stability within the homogeneous manifold only. We uncover the different attractors of the self-coupled system with the help of the bifurcation analysis software AUTO-07p [66], using the common external input $p \geq 0$ and the coupling strength $\epsilon \geq 0$ as control parameters. Despite both parameters being positive quantities, we also include negative values of $p$ in the bifurcation diagrams in order to reveal the entire bifurcation structure.

We start by recalling the dynamics of the uncoupled Jansen model (for details readers can refer to [63]). Fig 1(a) shows the value of the mean membrane potential of the PN population $v$ corresponding to different solutions of system (18) for fixed $\epsilon = 0$ and varying baseline input $p$. For small positive $p$, two stable steady states coexist (dark pink curves), each of them leading to a different type of stable periodic state (black curves) upon increasing the baseline input. First, the high-activity fixed point (upper pink branch) undergoes a supercritical Hopf bifurcation ($\text{HB}_1^+$) at $p \approx 90$, which corresponds to the onset of alpha oscillatory activity ($\sim$10Hz). This periodic state persists until $p \approx 315$, where it vanishes through a second supercritical Hopf-bifurcation ($\text{HB}_2^+$) leading again to a stable high-activity steady state. Second, the low-activity stationary state for $p$ small (lower dark pink branch) vanishes through a saddle-node in a invariant cycle (SNIC) bifurcation for $p \approx 114$, giving rise to finite amplitude oscillations at a theta range ($\sim$4Hz). This spiky oscillatory activity vanishes at $p \approx 137$ through a fold (or saddle-node) bifurcation of limit-cycles ($\text{FLC}_1$).

The addition of coupling ($\epsilon > 0$) modifies this bifurcation scenario. For instance, Fig 1(b) shows the bifurcation diagram for $\epsilon = 4$. The figure shows that coupling eliminates the supercritical Hopf bifurcation $\text{HB}_1^+$, so that for small values of $p$ only the low-activity fixed point is stable. As in the uncoupled case, this steady state vanishes at $p \approx 111$ through a SNIC bifurcation. However, now the branch of oscillatory states arising from the SNIC connects both the

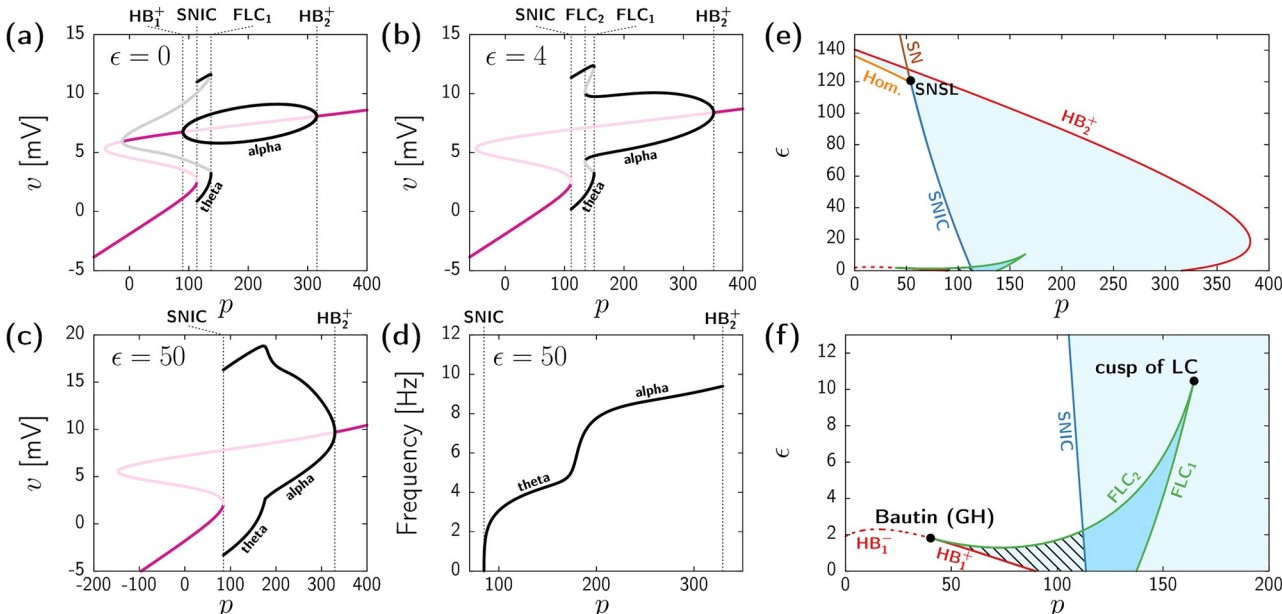

**Fig 1. Bifurcation diagrams of the homogeneous states subject to uniform perturbations.** (a-c) One-parameter bifurcation diagrams of Eq (18) obtained by varying $p$ and with fixed $\epsilon = 0$ (a), $\epsilon = 4$ (b), and $\epsilon = 50$ (c). Colored curves indicate stable fixed points (dark pink), unstable fixed points (light pink), extrema of stable limit-cycles (black), and extrema of unstable limit-cycles (grey). Dashed black vertical lines indicate the relevant bifurcation points cited in the text. (d) Frequency of the stable oscillatory state by varying $p$ and with fixed $\epsilon = 50$. (e,f) Two-parameter bifurcation diagram of Eq (18) depending on the external input $p$ and the coupling strength $\epsilon$. Panel (f) is a zoomed version of (e). Curves indicate different bifurcation types: supercritical Hopf (HB$^+$, continuous red), subcritical Hopf (HB$^-$, dashed red), saddle-node (SN, brown), saddle-node in a invariant cycle (SNIC, dark blue), saddle-node of limit-cycles (FLC, green), and homoclinic (Hom., orange) The light-blue region indicates the existence of a single periodic state. The light-blue region with stripped black pattern indicates the coexistence between a limit-cycle and a fixed point. The dark-blue region indicates the coexistence of two stable periodic states. All results obtained by analyzing the system of Eq (18) using the bifurcation analysis software AUTO-07p (scripts for one-parameter bifurcations available at www.github.com/pclus/transverse-instabilities).

theta ($111 \lesssim p \lesssim 150$) and alpha ($134 \lesssim p \lesssim 351$) frequency bands, which coexist in a small region bounded by two FLCs. Hence, for small but non-zero coupling the alpha oscillatory state emerges through the fold of cycles FLC$_2$ instead of a Hopf bifurcation, and, as in the uncoupled case, it disappears through the HB$_2^+$ supercritical Hopf bifurcation for large input ($p \approx 351$).

Further increase of $\epsilon$ leads to a total disappearance of bistability between oscillatory states. Fig 1(c) shows an example of this situation for $\epsilon = 50$. Upon increasing the external input $p$, the low activity fixed point leads to finite-amplitude oscillations through the SNIC bifurcation ($p \approx 84.68$). This oscillatory state becomes the only attractor for a wide range of $p$, until it vanishes at the supercritical Hopf bifurcation HB$_2^+$ ($p \approx 330$), beyond which trajectories converge to the high-activity fixed point. The frequency of this single stable oscillatory state (see Fig 1(d)) shows two distinct plateaus that dominate for most values of $p$, located at around the theta (3–5Hz) and alpha (7–9Hz) levels.

Overall, the diagrams in Fig 1(a) and (c) reveal a change on the onset of alpha oscillatory activity, together with a loss of bistability as $\epsilon$ increases. These changes can be better understood from the two-parameter bifurcation analysis shown in Fig 1(e) and (f). These diagrams illustrate the different regions of stability of Eq (18) for varying $p$ and $\epsilon$, with panel (f) showing a zoomed version of panel (e) for small coupling. The red continuous curve in Fig 1(e) and (f) corresponds to the Hopf bifurcation HB$_1^+$ giving raise to alpha activity for low values of $\epsilon$ (see Fig 1(a)). This bifurcation becomes subcritical (dashed red line) through a Bautin (or generalized Hopf) codimension-2 bifurcation at $p \approx 40$, $\epsilon \approx 1.8$. At this point, the FLC$_2$ appears

(green line), and joints the $FLC_1$ at a cusp ($p \approx 165$, $\epsilon \approx 10$) at which both bifurcations vanish. Therefore, within the dark-blue region delimited by the two FLCs (green curves) and the SNIC bifurcations (dark-blue curve) two stable oscillatory states coexist. Beyond this region, and for a wide range of $\epsilon$ values, the dynamical landscape of the system becomes simpler and coincides with that shown in Fig 1(c) (see Fig 1(e) for $\epsilon = 50$). This scenario changes for very high coupling, when the SNIC bifurcation turns to a SN through a saddle-node separatrix loop (SNSL) codimension-2 bifurcation [67]. From this point, a homoclinic bifurcation (Hom.) bounds the region of oscillatory dynamics, which appear for arbitrary low $p$. Parallel to the homoclinic line, two additional branches of FLC appear, leading to a very narrow region of oscillatory bistability. Since they have a minor effect on the overall bifurcation landscape, we do not depict them in Fig 1(e). Finally, further increase of $\epsilon$ ceases all oscillatory activity through the $HB_2^+$ (red continuous curve).

In summary, the dynamics of the system within the homogeneous invariant manifold can be divided in three main regions:

- Low activity fixed point, represented by the white region to the left of the SNIC bifurcation (dark blue curve) in Fig 1(e).

- High activity fixed point, represented by the white region to the right and above of the $HB_2^+$ line (red curve) in Fig 1(e).

- Oscillatory activity at the theta or alpha ranges (or bistability between both), mostly delimited by the $HB_2^+$ and the SNIC bifurcation curves (light blue shadowed region) in Fig 1(e).

Overall, the bifurcation analysis of Eq (18) uncovers the rich dynamical repertoire of the homogeneous manifold of the full system (Eq 10). Nonetheless, we remark again that this analysis only concerns homogeneous states, i.e., the different regions of stability and instability outlined so far only assume perturbations that act identically at each network node. In the next section we assess the stability of the homogeneous solutions to heterogeneous perturbations.

**Transverse stability.** The analysis of system (18) discussed above reveals the loci of all homogeneous states of the network model that are stable to uniform perturbations. In order to determine the stability of these states to arbitrary non-uniform perturbations, we follow a well-established approach that can be applied to either fixed points or periodic states (see Methods). This technique, analogous to the one used in the study of Turing bifurcations in complex networks [54, 68, 69], consists on decomposing an arbitrary perturbation vector on the basis given by the eigenvectors of a suitable matrix representing the way the nodes are coupled. This provides a dispersion relation for the growth rate of the perturbations. In our case, instead of the Laplacian matrix used in diffusively coupled systems, we diagonalize the normalized structural connectivity matrix $\tilde{W}$. As we will see, stable fixed points in the homogeneous manifold remain always stable to heterogeneous perturbations in the full-model. Hence, our focus will be on the stability of the limit-cycle solutions, for which this decomposition technique is known as the Master Stability Function (MSF) [58, 59, 61].

Fig 2(a) shows the first 5 eigenvectors of $\tilde{W}$ in terms of their components across the 90 brain regions. The first eigenmode is homogeneous, and its associated eigenvalue is always $\Lambda_1 = 1$ (see Methods). Perturbations along this direction are the ones already accounted by the analysis of Eq (18) in the previous section. The subsequent eigenmodes are heterogeneous and their associated eigenvalues are between -1 and 1 (see Methods). Perturbations along these directions are *transverse* to the homogeneous invariant manifold. Despite stemming from an irregular network topology, the eigenvectors exhibit a well defined spatial structure, which can be traced back to the well-known exponential decay of inter-region connectivity with

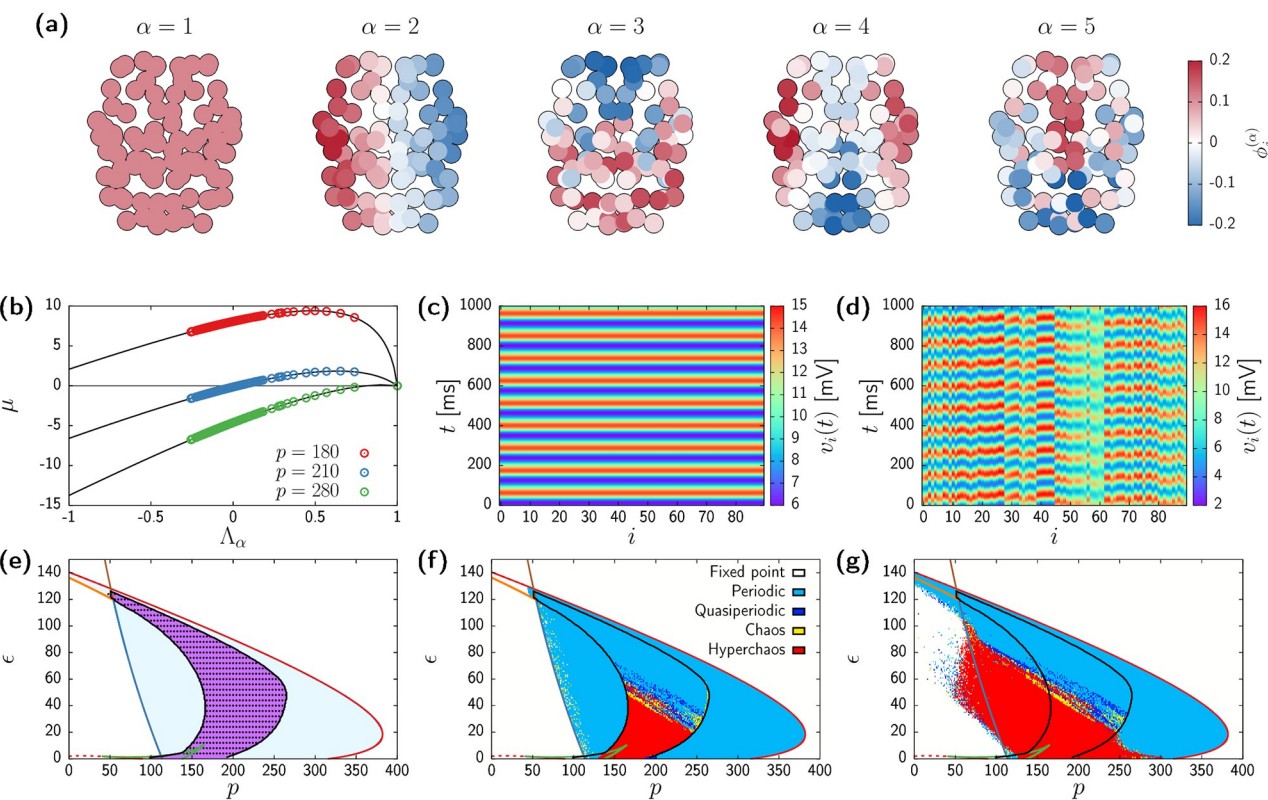

**Fig 2. Transverse instabilities of oscillatory states.** (a) First 5 eigenvectors of $\tilde{W}$ in a spatial representation of the brain (superior view). Each network node has been colored according to their contribution to the corresponding eigenvector $\Phi^{(\alpha)} = (\phi_1^{(\alpha)}, \ldots, \phi_N^{(\alpha)})$. (b) Master Stability Function of system (10) showing the dependence of the largest Floquet exponent $\mu$ with respect to the structural connectivity eigenvalues $\Lambda_\alpha$ for three different values of the input $p$. $\epsilon = 50$ in all cases. Circles correspond to the eigenvalues of $\tilde{W}$, whereas black curves are obtained by continuously tuning $\Lambda$. (c,d) Results from numerical simulations of Eq (10) with (c) $p = 280$ and $\epsilon = 50$ and (d) $p = 210$ and $\epsilon = 50$. (e) Complete bifurcation diagram of the homogeneous states. Colors of lines and regions as in Fig 1(e) and (f), with the region of transverse instability (i.e., $\mu_2 > 0$) shaded in pink and delimited with a black curve. Black circles correspond to numerical simulations in which $\langle \sigma \rangle > 10^{-5}$. Simulations initialized close to a homogeneous state. (f,g) Bifurcation diagram obtained from direct simulations of the system with initial conditions close to homogeneous (f) and random (g). Continuous curves as in Fig 1(e) and (f). Regions colored according the dynamical classification given by the two largest Lyapunov exponents (see Methods).

Euclidean distance observed in brain connectivity data obtained with diffusion tensor imaging (DTI) [70, 71].

Following the technique explained in Methods, we found that no transverse instabilities arise from the homogeneous fixed points. Therefore next we focus on the stability of the limit-cycle solutions by means of the Master Stability Function (MSF) [58, 59, 61]. The growth rate of an infinitesimal perturbation of a periodic state is given by the real part of the corresponding Floquet exponents [72], which we denote by $\mu$. The MSF provides the largest growth rate $\mu$ of a perturbation acting along each eigenmode $\Phi^{(\alpha)}$ as a function of its associated eigenvalue $\Lambda_\alpha$,

$$\mu_\alpha = \text{MSF}(\Lambda_\alpha) . \tag{1}$$

This relation is analogous to dispersion relations in spatially extended systems, where the growth rate of a perturbation is given as a function of the perturbation wavenumber [55]. In the Jansen model, the MSF needs to be computed numerically (see Methods section for a detailed explanation of its derivation and numerical computation).

Fig 2(b) shows the MSFs of the homogeneous limit-cycle solution for different $p$ values. Each circle indicates the real part of the largest Floquet exponent corresponding to a

perturbation applied along the $\alpha$th eigenvector computed according to the MSF. Since we are considering a stable limit-cycle solution given by the analysis of system (18), the largest growth rate corresponding to the uniform eigenvector (i.e., $\Lambda_1 = 1$) is $\mu_1 = 0$. The other exponents might be positive or negative depending on both the system parameters and $\Lambda_\alpha$. For instance, for $p = 280$ (green circles) the dispersion relation is negative for all the structural eigenvalues $-1 < \Lambda_\alpha < 1$. Therefore, in this case we expect small inhomogeneous perturbations of the homogeneous state to decay exponentially. In contrast, for $p = 210$ (blue circles) some of the connectivity matrix eigenvectors have positive growth rates $\mu_\alpha$, thus small perturbations should give rise to heterogeneous patterns. Indeed, Fig 2(c) and 2(d) show the results of integrating numerically Eq (10) for $p = 280$ and 210, respectively. The initial conditions correspond to a uniform state plus a small random perturbation. In agreement with the results given by the stability analysis, after a short transient (not shown), the dynamics for $p = 280$ falls back to the homogeneous state, whereas a heterogeneous spatiotemporal pattern arises for $p = 210$.

The MSF dispersion relations represented in Fig 2(b) show that the positive growth rates appear first through the second largest structural-connectivity eigenvalue, $\Lambda_2$. We thus extensively analyze the loci of unstable directions by checking whether $\mu_2$ has a positive real part for the entire region of existence of oscillatory activity. Fig 2(e) displays the region where $\mu_2 > 0$ (purple) superimposed on the bifurcation diagram of the homogeneous manifold, calculated in the previous section and shown originally in Fig 1(f). Remarkably, the region of transverse instability occupies a large portion of the parameter space. Moreover, the corresponding values of $\epsilon$ cover a range comparable to the other coupling parameters of the system, $C_1, \ldots, C_4 \in [0, 135]$.

In order to validate the emergence of heterogeneous dynamics in the system we perform numerical simulations of system (10) for different values of $p$ and $\epsilon$, starting from initial conditions close to a homogeneous state. At each time step we compute the standard deviation across network nodes,

$$\sigma(t) \coloneqq \frac{1}{N}\left[\sum_{i=1}^{N}(v_i(t) - \bar{v}(t))^2\right]^{\frac{1}{2}} \qquad \text{where} \qquad \bar{v}(t) = \frac{1}{N}\sum_{i=1}^{N}v_i(t) \; . \tag{2}$$

This quantity vanishes in the homogeneous states, whereas it is positive in heterogeneous states. Black circles in Fig 2(b) indicate the parameter values for which $\langle\sigma(t)\rangle > 10^{-5}$ in the simulations, showing a complete overlap with the results coming from the linear stability analysis (purple region in the figure).

Finally, we characterize the type of dynamical states arising in the region of transverse instability, by computing the two largest Lyapunov exponents (LE, see Methods), $\lambda_1$ and $\lambda_2$, in independent numerical simulations with varying $p$ and $\epsilon$. Using this tool we can classify the attractors of the system depending on the sign of the two exponents (see Methods). Fig 2(f) and 2(g) show the resulting numerical bifurcation diagrams corresponding to initial conditions close to a homogeneous state (Fig 2(f)) or entirely random initial conditions (Fig 2(g)). In both cases the region of transverse instability can be roughly divided in two parts: one dominated by periodic heterogeneous oscillations (large $\epsilon$, blue), and one displaying chaotic dynamics (small $\epsilon$, red) with at least two unstable directions. Additionally, simulations initialized at random (Fig 2(g)) show that the chaotic regime extends much beyond the transverse instability region, thus uncovering a coexistence region between spatiotemporal chaos and homogeneous states. In the next sections we investigate the properties of these different dynamical regimes in detail.

## Periodic travelling waves

The simplest instance of heterogeneous dynamics in system (10) is a periodic regime (blue region in Fig 2(f) and 2(g)). In this regime, the dynamics of each brain region is purely periodic, but with a non-zero phase difference between regions, i.e., there is phase-locking between nodes. Here we reveal that a multiple of such states might exist for a single choice of parameter values.

Fig 3(a) and 3(b) show the difference between each node's phase $\phi_j$ and the collective phase $\Psi$ (see Methods) for two simulations with fixed $p = 230$ and $\epsilon = 50$ and different initial conditions. In both cases the initial conditions are set close to the homogeneous (unstable) limit-cycle, plus a small random perturbation. The first case (simulation A, Fig 3(a)) reveals a wave pattern that travels from the left to the right hemisphere, whereas the second (simulation B, Fig 3(b)) displays a travelling wave that goes from the parietal to the frontal region, (see also S1 and S2 Movies).

Following [30] we obtain the direction and speed of the wave propagation by means of constrained natural element differentiation (see Methods, and also [30, 73]). The resulting vector field (Fig 3(c) and 3(d)) provides better indication on the specific type of pattern shown by each simulation, together with the wave propagation speed of each region. These propagation speeds are heterogeneous in space, as expected from the irregular brain connectome, with values ranging between 1 and 8m/s (see Fig 3(e)), in close agreement to results from non-invasive recordings, but one order of magnitude larger than those observed in invasive recordings [74]. This disagreement might be due to the fact that ours is a model of inter-cortical wave

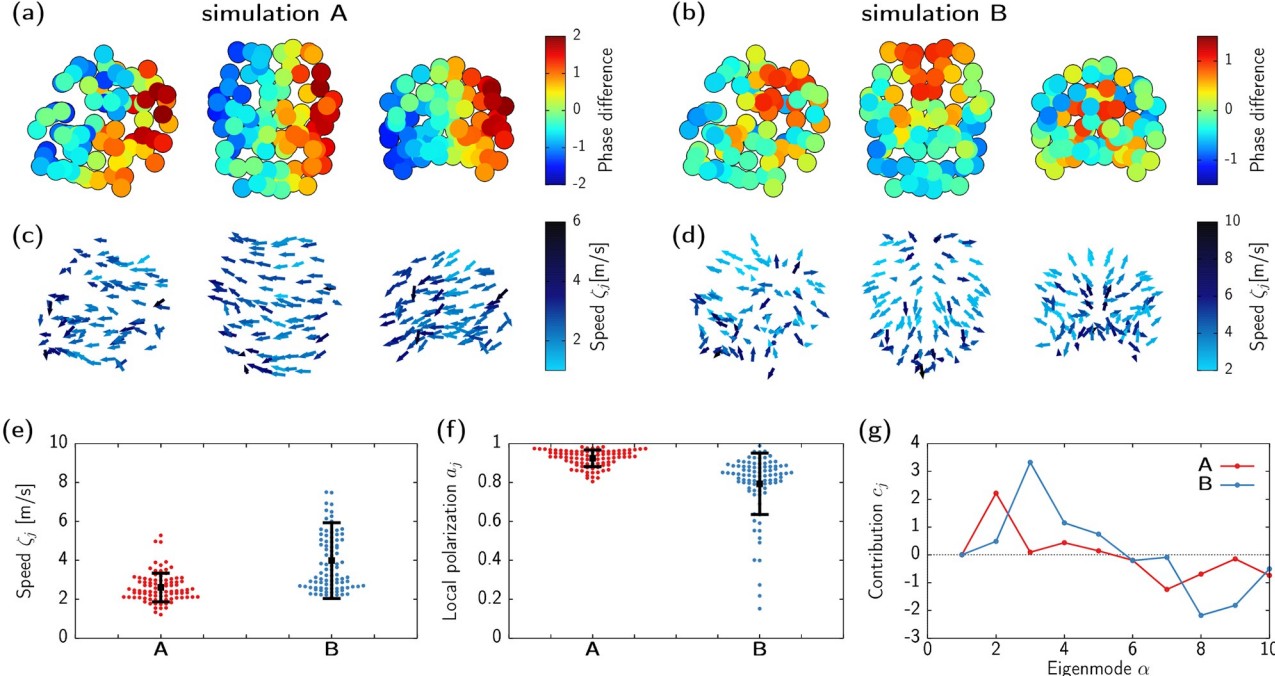

**Fig 3. Multistability between different types of periodic travelling waves.** (a,b) Difference between each node's phase $\phi_j$ and the collective phase $\Psi$ corresponding to simulations with $p = 230$ and $\epsilon = 50$, in the spatial representation of the brain network (from left to right: frontolateral, superior, and frontal views). Panels (a) and (b) correspond to two different initial conditions. (c,d) Propagation direction vectors $\zeta_j/\zeta_j$ corresponding to the phase patterns of (a) and (b). Color indicates the propagation speed $v_j$. (e,f) Swarm plot of propagation speed $\zeta_j$ (e) and local polarization $a_j$ (f). The circles correspond to individual brain regions. Black squares show the average over the entire network, and error bars indicate standard deviation. (g) Contribution of each structural connectivity eigenmode to the growth rate of the perturbations in simulations A and B (see Eq (29) in Methods).

propagation, whereas it has been argued that travelling waves in the scalp are mediated by an intra-cortical mechanism [74]. In fact, travelling waves in the cortex are usually observed not at the whole-brain level but localized in smaller regions [5, 75, 76]. A reconciling approach could consist on applying the same analysis in a model closer to experimental conditions, based on short-range coupled NMMs in a small section of the cortex.

Vector fields in Fig 3(c) and 3(d) show coherent spatial propagation patterns visible to the naked eye. Nonetheless, in the model, the spatial support is discrete and the coupling among nodes is set by a complex network, thus local irregularities or outliers should be expected. We quantify the local directional agreement among neighbouring propagation vectors by means of a measure of local polarization $a_j$ (see Eq (34) in Methods). Fig 3(f) shows the average (black squares) and individual (small circles) polarizations for simulations A and B. In both cases the different velocity vectors display a good agreement with the propagation directions of its neighbouring regions, as indicated by $a_j$ being close to 1. Nonetheless, simulation B presents a few outliers, which we have checked to correspond to parietal brain regions close to the source of the wave, where the pattern spreads in many different directions.

The well-structured wave patterns in Fig 3(a) resemble some of the eigenmodes given by the diagonalization of $\tilde{W}$ depicted in Fig 2(a). Indeed, we expect the eigenmodes associated to positive Floquet exponents in the MSF to have a larger role on the final state of the system. Nonetheless, the random perturbations used to initialize the system might be more localized along a specific direction, making that direction more prominent among other unstable modes. Fig 3(g) shows this specific contribution for simulations A and B as approximated by Eq (29) (Methods). The perturbation used in A is mostly being expanded through the second eigenvector (see Fig 2(a)), which shows a left-to-right division in close agreement to the observed pattern. Instead, perturbation B is mostly being expanded according to the third structural eigenvector, leading to the back-to-front wave propagation.

## Chaotic wave dynamics

Numerical analysis displayed in Fig 2(f) and 2(g) shows a large region of chaotic dynamics. Here we analyze an instance of such states. Lyapunov exponents are a common tool to characterize the complexity of a chaotic regime (see Methods) [77]. We set parameter values to $p = 170$ and $\epsilon = 20$, for which at least 5 Lyapunov exponents are positive, an indication of high-dimensional chaos. Single nodes still oscillate at the alpha range ($\sim$9Hz), however, the phase differences between brain areas change over time. Fig 4(a) shows consecutive time snapshots of the difference between the phase of each region $\phi_i$ and the collective phase $\Psi$, with the corresponding propagation direction vector field depicted in Fig 4(b). Such phase gradient preserves certain levels of locally coherent structure, although the scenario looks much more complex than the periodic travelling waves presented in the previous section (see also S3 Movie).

In order to understand the main features of these chaotic patterns, we first analyze the level of phase synchrony in the network. By observing the time evolution of the phase differences (see Fig 4(c)), we can appreciate that the dynamics alternates between periods in which all nodes evolve with similar phases (all phase differences are distributed around zero), with periods in which a group of nodes (but not all) lose synchrony. Such type of breathing dynamics can be monitored by means of the Kuramoto order parameter $R$ (see Methods for a definition). This can be seen in Fig 4(d), which shows the irregular oscillatory activity of $R$. These results indicate that the level of collective synchronization in the network itself displays chaotic behavior. The regularity of both the phase differences and $R$ can be quantitatively analyzed by means of their corresponding autocorrelation functions (Fig 4(e)). The quick decay of both

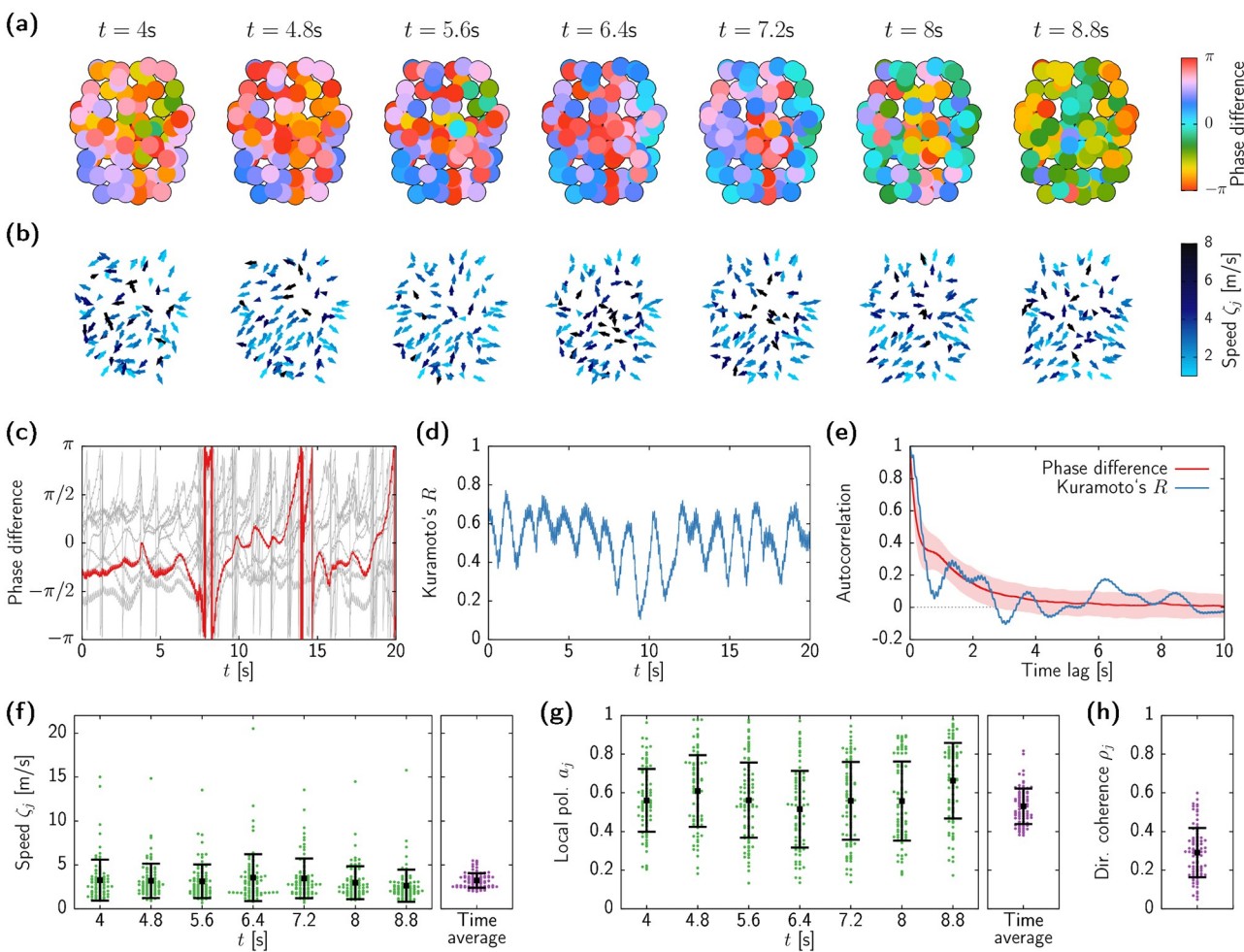

**Fig 4. Spatiotemporal dynamics of the chaotic state.** (a) Snapshots of the phase differences $\phi_j - \Psi$ between each node's phase and the collective phase for a simulation with $p = 170$, $\epsilon = 20$. (b) Snapshots of the propagation directions colored according to the corresponding speed. (c) Time series of the phase difference $\phi_j - \Psi$ for 10 nodes (grey curves), with an individual brain region highlighted with a red thick line. (d) Time evolution of the Kuramoto order parameter $R$. (e) Autocorrelation of $R$ (blue curve) and average correlation of the phase differences (red curve), with the corresponding standard deviation shown as a light red shadow. (f-h) Swarm plot of instantaneous propagation speeds $\zeta_j$ (f), local polarization $a_j$ (g), and directional coherence $\rho_j$ (h), corresponding to the snapshots in (a). Circles show the values of individual brain regions, black squares indicate average value, and error bars show standard deviation.

autocorrelations illustrates the short-time scale of the dynamics, which agrees with that indicated by the inverse of the largest Lyapunov exponent ($1/\lambda_1 \simeq 0.93$s). Hence, overall, the synchronization patterns in the network, either pairwise or collective, are mediated by the spontaneous appearance of a slow modulation of the alpha activity.

We next characterize this chaotic wave propagation using different measures. Fig 4(f) shows the average and standard deviation of the propagation speed for each node. The velocity range is similar to that of the periodic waves (1–8m/s), with a few outliers with larger velocities. These outliers are not present on the time-averaged distribution (right panel in Fig 4(f)), thus indicating that they correspond to sporadic fluctuations. The local polarization (Fig 4(g)) exhibits a similar behavior: Instantaneous snapshots of $a_j$ display a large variability on the phase coherence across regions, but with a time average concentrated around $\langle a_j \rangle \approx 0.5$ for all nodes. Finally, we quantify the change of the propagation direction over time using the mean resultant length $\rho_j = \| \langle \zeta_j / \zeta_j \rangle \|$ (see Methods), which measures the directional coherence of

single nodes, being 0 for completely random directions, and 1 for quenched directionality (as in periodic travelling waves). The low values of $\rho_j$ in Fig 4(h) indicate that most nodes propagate their dynamics to statistically almost any direction over the entire simulation. Altogether, this is a manifestation of short-lived and locally coherent wave dynamics, which travel through the network without a well-identified pattern.

## Onset and multistability of irregular dynamics

In this section we characterize the emergence the spatiotemporally chaotic dynamics displayed by our large-scale brain model, as well the effect of transverse instabilities on the frequency of the oscillations. We perform different sets of numerical simulations for fixed $\epsilon = 50$ and varying $p$, starting from random initial conditions. Fig 5(a) shows the 5 largest Lyapunov exponents of the system as $p$ is varied (colored thin lines, left axis). The transverse instability takes place at $p \simeq 265.5$ (right-most vertical dashed line in the figure), and the first instances of chaos emerge for values of $p$ smaller than $p \simeq 184.5$ (first vertical dashed line in Fig 5(a)). Between these two values, the behavior of the system is either periodic (single zero LE) or quasiperiodic (two LEs equal to zero). Chaotic dynamics appear rather abruptly from the quasiperiodic states as $p$ is decreased.

We therefore conclude that the route to chaos is that of a torus breakdown [78, 79]. Examples of periodic, quasiperiodic, and chaotic dynamics are displayed in Fig 5(b). In this panel, each plot shows the time series of the sample mean voltage, $\bar{v} := \frac{1}{N}\sum_{j=i}^{N} v_i$, corresponding to simulations with $p = 180$ (top), 177 (middle), and 175 (bottom). The quasiperiodic state (middle row of Fig 5(b)) presents an additional slow frequency on top of the alpha activity, which

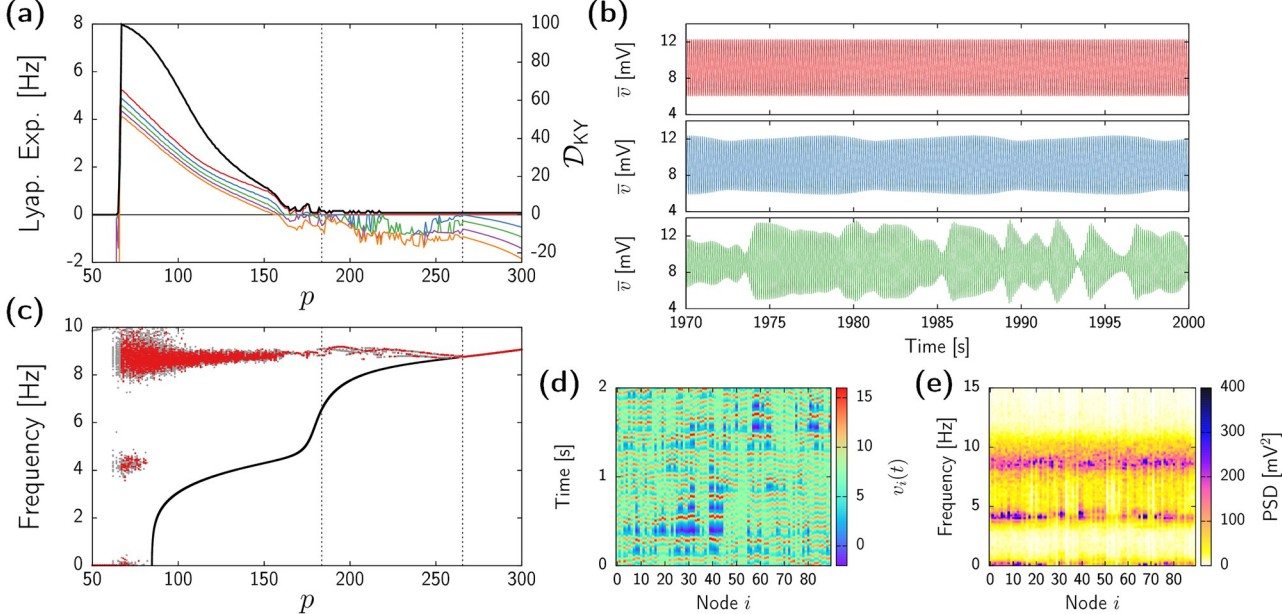

**Fig 5. Onset of chaotic dynamics and multistable dynamics.** (a) Five largest Lyapunov exponents (thin coloured lines, left axis) and Kaplan-Yorke dimension $\mathcal{D}_{KY}$ (black thick line, right axis) for $\epsilon = 50$ and varying $p$, as obtained from direct integration of system (10). Simulations at each value of $p$ were initialized at random. Vertical dashed lines indicate the onset of transverse instability (right, $p \simeq 265.5$) and the onset of chaos (center, $p \simeq 184.5$). (b) Time series of the mean-field voltage of the pyramidal neurons, $\bar{v}$, for $p = 180$ (top), 177 (middle), and 175 (bottom) and fixed $\epsilon = 50$. (c) Peak frequency of each network node. The black line shows the oscillation frequency of the underlying homogeneous state, as in Fig 1(f). Dots indicate the frequency for which the power spectral density is maximal for each node. Results from 5 simulations superimposed in gray, with one replicate highlighted in red. (d) Activity pattern of each node obtained from a simulation with $p = 70$ and $\epsilon = 50$. Only a time span of 2 seconds shown. (e) Power spectral density of each network node as obtained from the same simulation as in (d).

then breaks into faster irregular modulations of the alpha rhythm at the chaotic state (bottom row of Fig 5(b)).

After the onset of chaotic dynamics at $p \simeq 265.5$, further decrease of $p$ leads to consecutive Lyapunov exponents becoming positive, thus increasing the dimensionality of the chaotic attractor. In fact, although we only display 5 Lyapunov exponents, many more directions become unstable. We estimate the (fractal) dimension of the chaotic attractor in terms of the Kaplan-Yorke dimension, $\mathcal{D}_{KY}$ [77, 80] (see Methods), which quantifies the effective number of degrees of freedom that characterizes the irregular dynamics (black thick line in Fig 5(a), right axis). This analysis reveals the high-dimensionality of the chaotic regime, which can reach up to 100 for low $p$ values. For even lower $p$ the complex oscillatory state suddenly vanishes, and the dynamics converge to the stable homogeneous fixed point.

Does the onset of heterogeneous states modify the oscillatory frequency of the system? To answer this question we analyze the leading frequency of each network node upon varying $p$. The result is shown in Fig 5(c), where dots indicate the peak frequency of each brain area, as determined by the maximum of the power spectral density in each case. For each value of $p$, results from 5 different simulations are shown in gray, with results from a single randomly chosen simulation depicted in red. Before the synchronous alpha activity loses stability (right-most vertical dashed line in Fig 5(c), $p > 265.5$), all nodes in all simulations evolve according to the frequency given by the analysis of the self-coupled system (18), obtained in Fig 1(d) and indicated by a black curve. Between the transverse instability of the homogeneous state and the emergence of chaos ($184.5 \leq p \leq 265.5$, area between the two vertical dashed lines in the figure), all brain areas in a given simulation oscillate with the same leading frequency, even though different sets of simulations fall in different attractors with slightly different oscillation frequencies (due to multistability between different types of periodic states, as at the case analyzed in Fig 3). In the chaotic regime ($p \leq 184.5$) this situation changes, and within each simulation different nodes might oscillate at different peak frequencies, which cover a broader range as the chaotic state increases dimensionality. In any case, it is worth highlighting that oscillatory activity remains at the alpha range, around 9Hz, for most values of $p$. This frequency range differs from that of the underlying homogeneous state, which transitions to the theta regime and finally vanishes (see black curve).

As shown in Fig 5(c), vestiges of theta activity exist only for values of $p$ close to the steady-state regime (e.g., $p \simeq 70$). In those cases, brain regions oscillate intermittently between the alpha and theta frequency bands. Fig 5(d) shows an example of such multifrequency dynamics for $p = 70$ and $\epsilon = 50$, with the corresponding power spectra displayed in Fig 5(e). Similar multifrequency dynamics are usually associated to the role of stochastic input inducing jumps between the two oscillatory states of the uncoupled Jansen model [81, 82]. Here instead, it is the collective deterministic chaos what causes a two-peak spectrum for single nodes.

## Non-normalized connectivity

All results shown so far stem from the row-normalized connectivity matrix $\tilde{W}$ given by Eq (14), which allows for a semi-analytical treatment of the model. Next, we turn to a numerical analysis of the model using the non-normalized network connectivity $W$ directly as obtained from the data. In this case one cannot ensure the existence of a homogeneous manifold, and thus the fixed points and their stability should be assessed by solving a system of 540 nonlinear equations and analyzing the corresponding Jacobian. Here we avoid such a full stability analysis of the system, since direct simulations are enough to compare with our previous results.

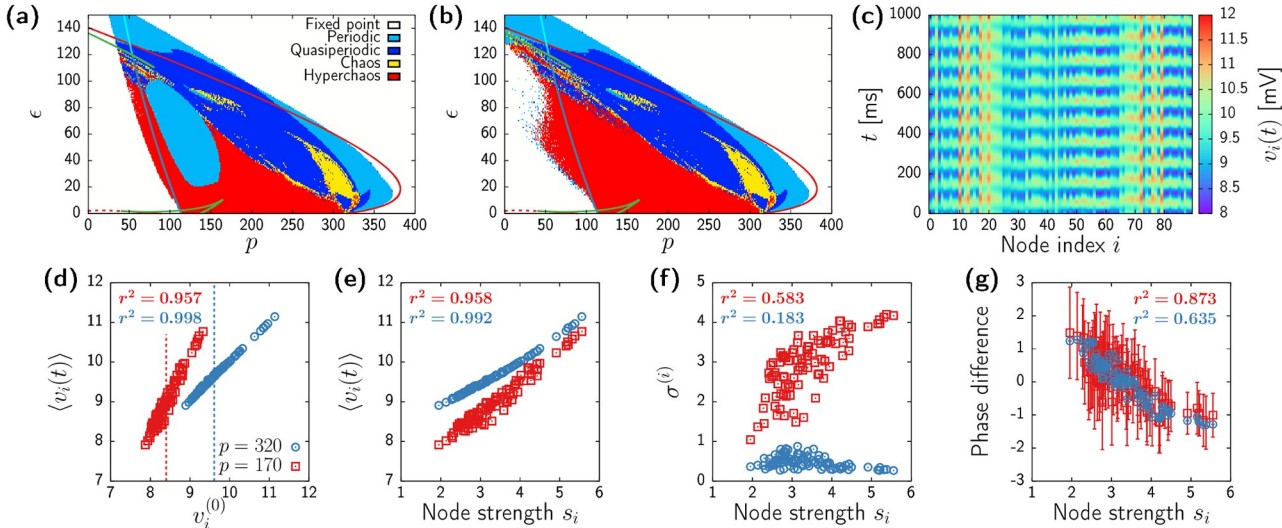

**Fig 6. Dynamics of the Jansen model with unnormalized connectivity.** (a) Numerical bifurcation diagram as obtained from the Lyapunov analysis of numerical simulations. Initial conditions set close to homogeneous. Continuous lines identical to those of Fig 1(f). (b) Same than (a), but initial conditions set at random. (c) Time evolution of each network node for $p = 320$ and $\epsilon = 50$ starting with random initial conditions. (d,e,f,g) Relation between different measures obtained from numerical simulations for $\epsilon = 50$ and $p = 320$ (red circles) and $p = 170$ (blue squares). The corresponding squared Pearson's correlation coefficients $r^2$ are displayed in each plot. (d) Correlation between oscillation focus $\langle v^{(i)} \rangle$ and the underlying unstable fixed point $v_0^{(i)}$. Dashed vertical lines correspond to the homogeneous fixed points of the normalized system $\tilde{W}$. (e) Correlation between oscillation focus $\langle v^{(i)} \rangle$ and node strength $s_i$. (f) Correlation between oscillation amplitude $\sigma^{(i)}$ and node strength $s_i$. (g) Correlation between the phase difference $\phi_i - \Psi$ and node strength $s_i$. Error bars correspond to temporal standard deviation.

We analyze numerically Eq (10) using the SC matrix $W$. Therefore, we replace the input Eq (11) in Eq (10) by

$$I_i(t) = p + \frac{\epsilon}{\langle s \rangle} \sum_{j=1}^{N} w_{ij} \, \mathrm{Sigm}[v_j(t)] \tag{3}$$

where $\langle s \rangle$ is the average node strength, i.e.,

$$\langle s \rangle := \frac{1}{N} \sum_{i=1}^{N} s_i = \frac{1}{N} \sum_{i=1}^{N} \sum_{j=1}^{N} w_{ij}. \tag{4}$$

The scaling factor $\langle s \rangle^{-1}$ ensures that the distribution of incoming connections is centered at 1, so that the effect of the coupling strength $\epsilon$ in the non-normalized model is comparable to that the row-normalized case studied so far.

Fig 6(a) and 6(b) show the classification of dynamical states as obtained from the two largest Lyapunov exponents in numerical simulations. The first figure shows results obtained initializing the system close to a homogeneous state, whereas the second corresponds to random initial conditions. Continuous lines show the bifurcations of Eq (18) as in Fig 2. The diagrams reveal a scenario very similar to that of the normalized topology, with a central large region displaying different types of oscillatory states bounded by two white regions corresponding to a low-activity (left) and high-activity (right) fixed points. Also, the presence of hyperchaos (red region in the plots) is ubiquitous, and shares a small island of bistability with periodic oscillatory states which contrasts with the large bistable areas present in the normalized system (Fig 2). Additionally, the non-normalized system displays a large region of quasiperiodicity and a small island of low-dimensional chaos not present in Fig 2.

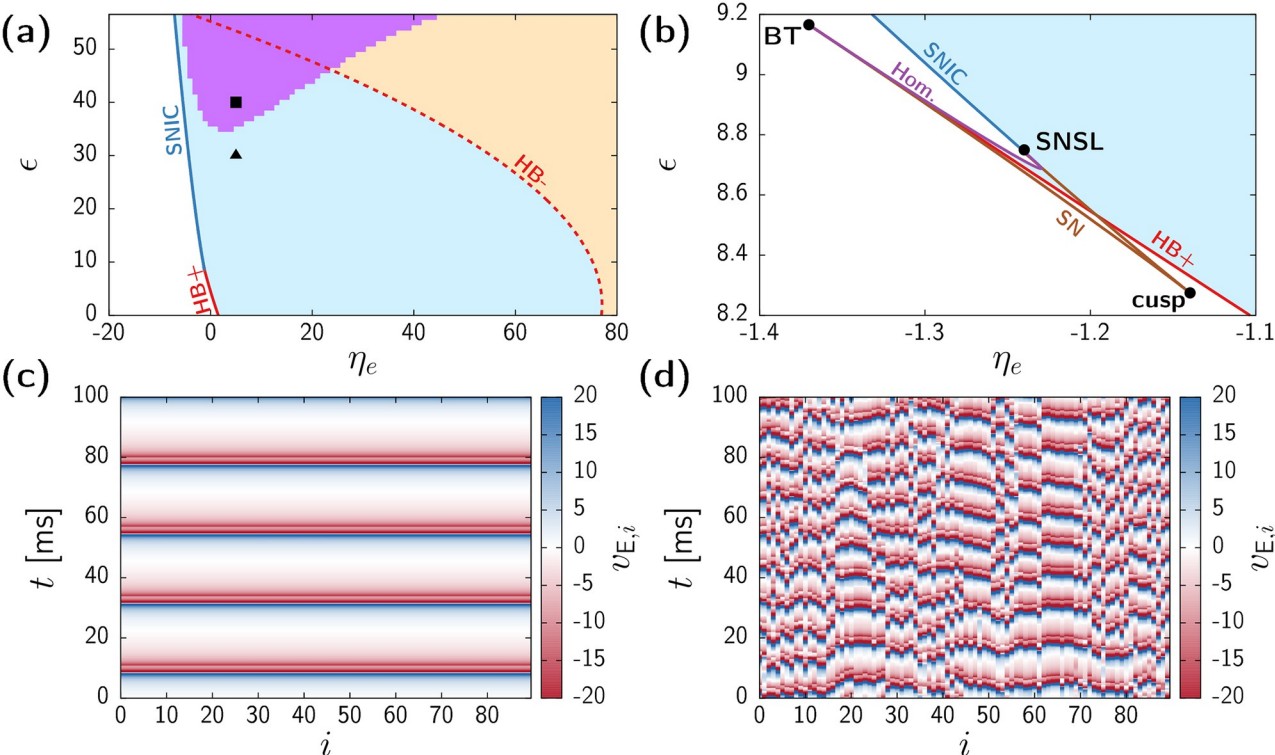

**Fig 7. Dynamics of the large-scale brain model composed of NG-NMMs.** (a,b) Complete bifurcation diagram of system (36). Panel (b) is a zoomed version of (a). Continuous lines correspond to the bifurcations of the homogeneous manifold given by Eq (38) obtained with AUTO-07p: saddle-node on an invariant cycle (SNIC, dark blue), supercritical Hopf (HB+, continuous red), subcritical Hopf (HB-, dashed red), saddle-node (SN, brown), and homoclinic (Hom., purple). Light blue shaded region indicates stability of the homogeneous gamma activity. Orange shaded region indicates bistability between homogeneous gamma activity and a fixed point. Pink shaded region indicates the region of transverse instability of the synchronized oscillations. Black triangle and square in panel (a) indicate the parameter values corresponding to panels (c) and (d) respectively. (c,d) Time evolution of each node as obtained from a numerical simulation of Eq (36) for $\epsilon = 30$ (c) and $\epsilon = 40$ (d). In both cases $\eta_e = 5$.

Despite the quantitative dissimilarities between the dynamical landscapes of the row-normalized and non-normalized systems described above, there is a fundamental difference between using $W$ or $\tilde{W}$ with regard the intrinsic dynamics of the system: In the non-normalized system all states are always heterogeneous. Fig 7(c), for instance, shows a simulation for parameter values close to the onset of oscillations ($p = 320$ and $\epsilon = 50$).

This oscillatory state arises from an underlying unstable heterogeneous fixed point that we obtain by solving the nonlinear fixed point equations of the system (see Methods). Indeed, Fig 6(d) shows that the time averaged mean membrane potential $\langle v_i(t) \rangle$, is practically identical to the underlying steady state $v_i^{(0)}$ for large enough $p$ (blue circles). In a chaotic state ($p = 170$, $\epsilon = 50$, red squares in Fig 6(d)) the center of the oscillations shifts from the fixed point, but the two quantities still show a high degree of correlation. Remarkably, the heterogeneous unstable fixed points remain distributed around the homogeneous steady state of the model with normalized connectivity for same parameter values (vertical dashed lines in Fig 6(d)). Hence, the use of the matrix $W$ does not seem to fundamentally alter the nature of the different steady states.

Next we study to what extent these oscillatory states depend on the distribution of inputs $s_i$. Fig 6(e), 6(f) and 6(g) show the focus, amplitude, and phase difference of each node plotted against the node strength for the periodic (blue circles) and chaotic (red squares) cases. First, the focus of the oscillations follows a direct relation with the input of each node (Fig 6(e)), for

both the periodic and chaotic dynamics. Second, the amplitude correlates mildly with the node strength at the chaotic region (red squares in Fig 6(f)), but appears independent from $s_i$ close to the onset of oscillatory activity (blue circles in Fig 6(f)) Finally, Fig 6(g) shows a negative correlation between the phase difference of each brain region and their corresponding node strength: Regions with lower input tend to oscillate first, and the hubs tend to be the last. In the periodic regime (e.g., $p = 320$) this relation translates to a travelling wave in which oscillations spread from outer to the inner region of the brain (see S4 Movie). In the chaotic regime ($p = 170$) the role of the node strength on the dynamics is still prominent (blue squares in Fig 6(g)), although much blurred by the irregular behavior of the system, as shown by the large error bars (see also S5 Movie).

Overall, the non-normalized network topology adds a new layer of complexity in the model, as homogeneous states no longer exist. Nonetheless, many features of the dynamics can be traced back directly to the distribution of node strengths $s_i$. Moreover, the agreement between diagrams in Fig 6(a) and 6(b) and those of Fig 2(f) and 2(g) indicate that the onset of chaotic dynamics is mostly retained in the row-normalized network topology $\tilde{W}$.

## Spatiotemporal chaos in a large-scale brain model with next generation NMMs

We have seen so far that irregular spatiotemporal dynamics arise in networks of coupled NMMs from transverse instabilities of oscillatory states. We expect this to be a general mechanism in brain networks. In order to illustrate this ubiquity, we now analyze a large-scale brain model composed of coupled next generation NMM (NG-NMM). These models are derived from an exact mean-field theory for quadratic integrate-and-fire neurons, and therefore, the corresponding firing rate equations can be traced back to the dynamics of single neurons [17]. Here we consider a pyramidal-interneuronal network gamma (PING) setup, in which the local dynamics within each brain region produces gamma activity through the interplay between excitatory and inhibitory neurons (see Methods, and also [83]). In order to obtain a bifurcation diagram, we proceed as we have done above for the Jansen system: first we investigate the homogeneous manifold of the system, and then apply the MSF formalism.

The manifold of homogeneous trajectories corresponds again to a self-coupled version of the model (Eq (38) in Methods). Fig 7(a) shows the bifurcation diagram corresponding to homogeneous trajectories of the system, using as control parameters the external-to-excitatory baseline input $\eta_e$ and the coupling strength $\epsilon$. For weak coupling ($\epsilon \ll 1$) and low $\eta_e$ the system remains in a single low-activity fixed point (white area in the plot) corresponding to a state of asynchronous dynamics. Upon increasing the constant external input $\eta_e$, such steady state gives raise to fast oscillatory activity (blue-shaded area) through a supercritical Hopf bifurcation (red continuous line). For large values of $\eta_e$ the fixed point recovers stability through a subcritical Hopf (dashed red curve), giving raise to a bistable state between gamma activity and asynchrony (orange-shaded area). For even larger values of $\eta_e$ gamma activity finally vanishes through a saddle-node of limit cycles (outside figure range).

As it happened with the Jansen system, increasing $\epsilon$ leads to a change on the onset of oscillatory activity. Following a typical scenario in oscillatory systems (see, e.g., [84–88]), in a tiny region of the parameter space (see panel (b)) three codimension-2 bifurcations coexist: a Bogdanov-Takens (BT), a saddle-node separatrix loop (SNSL), and a cusp of saddle nodes. The Hopf bifurcation vanishes at a Bogdanov-Takens (BT) point, and a saddle-node separatrix loop (SNSL) gives raise to a SNIC branch (dark blue line). Therefore, for most values of $\epsilon$, gamma activity arises through a infinite-period (SNIC) bifurcation. Also, an increase of the coupling causes the region of bistability between the oscillatory states and the fixed-point to

increase (orange-shaded area). Overall, homogeneous oscillatory activity –with or without bistability– dominates the bifurcation diagram. An example of such homogeneous gamma activity is displayed in Fig 7(c), corresponding to $\eta_e = 5$ and $\epsilon = 30$ (black triangle in Fig 7(a)).

By applying the MSF formalism on these oscillatory states, we unveil a region of transverse instability (pink-shaded region in Fig 7(a)), which emerges for large values of $\epsilon$. As a result, simulations initialized close to the homogeneous state within this parameter region exhibit irregular spatiotemporal patterns, as the one depicted in Fig 7(d) ($\eta_e = 5$, $\epsilon = 40$ i.e, black square in Fig 7(a)). Notice that the frequency of the oscillations becomes almost twice as that of the (unstable) homogeneous state, which in this case is approximately 46Hz. This is a differentiating feature with respect to the Jansen model, in which heterogeneous activity is faster than that of the underlying homogeneous state, but always close to the typical values displayed by the model.

Overall, the large-scale brain model with NG-NMMs displays a mechanism towards the onset of irregular states analogous to the Jansen case, despite the different nature of the two models [89]. For instance, we did not include synaptic dynamics in Eq (36), i.e., neurons receive instantaneous delta-like pulses. The onset of spatiotemporal chaos shown here supports thus the generality of transverse instabilitites in the oscillatory dynamics of coupled NMMs.

## Discussion

The idea that systems composed of simple deterministic subunits can give rise to complex spatiotemporal behavior traces back to the seminal work of Alan Turing [51]. That work showed that a homogeneous equilibrium in a system of diffusively coupled units might lose stability due to the coupling between neighbouring sites, thereby producing spatial patterns. Decades of research have extended this simple yet powerful framework to cover a wide range of possibilities, including instabilities arising from uniform oscillatory states [52, 53, 90] and pattern formation in complex networks [54]. Both of these extensions are embodied in the field of collective synchronization, in which the Master Stability Function provides a proper formalism to analyze the stability of homogeneous oscillatory states [58–62]. In brain networks, the stability of synchronized states has been analyzed only in simplified network topologies [48, 50] or by means of phase-reduction approximations that only apply for weak coupling [33, 35]. In this paper we have studied a general scenario, that reveals transverse instabilities of homogeneous states as an ubiquitous mechanism for the onset of travelling waves and high-dimensional chaos in large-scale brain models.

In computational neuroscience, the spontaneous emergence of patterns through instabilities of a uniform state has been emphasized mostly in the context of neural fields [91–95]. Neural fields are models defined in a continuous spatial support, where the synaptic coupling is a smooth function of the distance between regions. Hence, these type of models cannot capture the fine macroscopic organization of the brain connectome represented by complex networks, and are thus more adequate to model local intra-cortical dynamics [96, 97]. Nonetheless, the assessment of transverse instabilities is general enough to apply to both continuous spatial support with simplified interaction rules and neural mass models interacting through complex networks. The main difference between the two cases lays on the decomposition of the perturbation vector. In neural fields and regular network topologies [50], as in the Turing framework, stability analysis of homogeneous states is attained by decomposing a spatial perturbation in Fourier space. Instead, in complex networks composed of coupled NMM, the MSF requires the diagonalization of the structural connectivity matrix. Interestingly, some studies show that spectral analysis of whole-brain networks enables the characterization of

functional and resting brain activity [98–100]. Our work provides thus a mathematical framework to further explore the relation between structural connectivity (in terms e.g. of graph spectra) and functional activity data.

An important difference between large-scale neural dynamics and classical pattern formation lies in the nature of coupling. Classical pattern forming systems usually involve diffusive coupling, for which the homogeneous manifold of the system coincides with the dynamics of the uncoupled units [55]. In contrast, the long-range brain connections represented by structural connectivity matrices correspond to myelinated nerve tracts across brain regions. Hence, the interactions between nodes are mediated by chemical synapses driven by the firing rate of pre-synaptic regions. As a result, the homogeneous manifold of the system is given by a self-coupled version of the single-node model, and therefore the coupling strength modifies the uniform states of the system in a non-trivial manner. We have shown, for instance, that in both Jansen and next generation NMMs, there is a change on the onset of synchronized activity from a Hopf to a SNIC bifurcation upon increasing the coupling parameter $\epsilon$. Additionally, coupling eliminates all forms of bistability in the homogeneous states of the Jansen model, whereas it enlarges the region of coexistence between homogeneously attracting limit-cycles and steady states in the NG-NMM.

Most of the theory of pattern formation is based on studying instabilities from fixed points. Although we also analyzed this case, we did not find instabilities arising from stationary states. Instead, the linear stability of oscillatory solutions revealed the ubiquity of transverse instabilities as a route for the emergence of spatiotemporal chaos in large-scale brain models. This mechanism explains the onset of deterministic chaos outlined in previous works by means of direct numerical simulations [44, 49]. Moreover, the numerically computed Master Stability Functions (Fig 2(b)) show that the onset of unstable modes occurs through eigenvalues arbitrarily close to the zero eigenmode. This scenario is very close to that of the Benjamin-Feir instability in the Ginzburg-Landau system, which was studied by Kuramoto as a main route to turbulence in chemical reaction systems [52, 53, 60].

The emergence of high-dimensional chaos in large-scale brain networks is in line with recent studies that highlight the prominence of turbulent dynamics as a signature of healthy brain activity [8, 71, 101–103]. These complex fluctuations are observed in blood-oxygen-level-dependent (BOLD) signals obtained from fMRI scans, which are characterized by slow (below 1Hz) fluctuations. These are in the same time scale as the chaotic slow modulations of alpha rhythms that we observed in our brain network composed of Jansen NMMs. Remarkably, experimental studies have also found a negative correlation between the amplitude of alpha rhythms and BOLD activity [104–107]. Thus, we conjecture that the chaotic dynamics generated by transverse instabilities might capture the spatiotemporal fluctuations of BOLD signals observed in recordings.

Another important feature of the chaotic dynamics in the Jansen model is the coexistence of two frequency ranges, theta (~5Hz) and alpha (~10Hz), in some regions of the parameter space (see Fig 4(c)–4(e)). These alternating bursts of activity between the two rhythms are consistent with the multifrequency and transient behavior of oscillations in EEG recordings.

Large-scale brain models aim to trace the basic principles behind neural activity. Therefore, one might need to account for other sources of complexity not included in this work. For instance, despite being a common choice in the literature [30, 33, 35, 44, 45, 56, 57], in some cases it would be preferable to use non-normalized topologies. We have shown that regardless of this simplification, complex spatiotemporal patterns and high-dimensional chaos exist already in the row-normalized system. Moreover, the numerically-derived bifurcation diagram of the non-normalized system shown in Fig 6(a) and 6(b) qualitatively matches the dynamical landscape uncovered by the analysis of the homogeneous states of the simplified model.

Additionally, here we have considered long-range excitatory coupling targeting only pyramidal neurons. The MSF formalism can also be applied in the case where long-range excitation targets both excitatory and inhibitory units. A simple exploration of this setup using Jansen NMMs indicated that no transverse instabilities emerged in this situation (results not shown). This is in agreement with [50], which shows that self-sustained traveling waves emerge in a ring of Wilson-Cowan units when cross-excitation only affects excitatory populations but vanish if interneurons are also targeted.

Finally, the assumption of homogeneous dynamics across brain regions might be challenged by previous findings that indicate hierarchical heterogeneities in synaptic strengths and time scales [108, 109]. Nonetheless, a deeper understanding of transverse instabilities in simplified systems provides a solid ground to analyze the emergence and propagation of spatiotemporal neural activity in comprehensive models including such heterogeneities. To this end, future work should consider quantitative comparisons between the dynamics emerging from transverse instabilities and dynamical data from EEG or fMRI recordings. This might help to validate some of the aforementioned modeling choices as well as providing relevant estimations for $\epsilon$, usually given by fitting the model to available data (see, e.g. [8, 29]).

## Materials and methods

### Structural connectivity data

The structural connectivity of our NMM network has been obtained from diffusion tensor imaging (DTI) data of 16 healthy subjects, collected and analyzed in a previous study [65]. The different brain regions were defined using the AAL90 parcellation [64]. A $90 \times 90$ structural connectivity matrix $\mathbf{W}$ was obtained averaging the connectome of the 16 subjects. For more details about data collection and prepossessing we refer the readers to [65]. For details on the further normalization used in most of this paper, see Section entitled "Normalized connectivity" below.

### The Jansen NMM model

**Single region.** Jansen's NMM (also known as Jansen-Rit model) describes the dynamical activity of a cortical column of neurons [15, 16, 63]. Following principles derived from earlier works [13, 110], the model assumes populations of pyramidal neurons (PN) and inhibitory interneurons (IN), Recurrent connections are only present in the PN population, whereas inhibitory neurons solely receive inputs from pyramidal neurons For simplicity the model assumes that recurrent connections within the PN population are mediated through neurons that do not receive direct input from inhibitory neurons, and can therefore be interpreted as a third independent population, which we call recurrent pyramidal neurons (rPN). Finally, pyramidal neurons also receive excitatory external stimuli from other brain regions, modelled through a firing rate variable $I(t)$.

In the Jansen model the excitatory and inhibitory post-synaptic potentials (PSP) are given by

$$
\begin{aligned}
h_{\mathrm{e}}(t) &= Aate^{-at}H(t) \\
h_{\mathrm{i}}(t) &= Bbte^{-bt}H(t)
\end{aligned}
\tag{5}
$$

where $H$ is the Heaviside step function, $A$ and $B$ are the PSP amplitudes, and $a$ and $b$ quantify the synaptic time scales. As a result, a neural population receiving an excitatory firing rate of $r(t)$ generates an excitatory post-synaptic potential (ePSP) described by the second-order

differential equation

$$\ddot{y}(t) = Aar(t) - 2a\dot{y}(t) - a^2 y(t) \ . \tag{6}$$

Analogously, a inhibitory firing rate generates a inhibitory post-synaptic potential (iPSP) according to

$$\ddot{y}(t) = Bbr(t) - 2b\dot{y}(t) - b^2 y(t) \ . \tag{7}$$

On the other hand, a population with mean membrane potential $v(t)$ generates a firing rate according to the sigmoid function

$$\mathrm{Sigm}(v) \coloneqq \frac{2e_0}{1 + e^{r(v_0 - v)}} \ . \tag{8}$$

The model assumes that the self-dynamics of neurons within a population can be averaged out. Thus, the entire system evolves as determined by the evolution of the average PSPs of the different populations given by Eqs (6) and (7). As a result, the interactions between the three populations (PN, IN, and rPN) lead to the 6-dimensional Jansen model:

$$
\begin{aligned}
\dot{y}_0(t) &= y_3(t) \\
\dot{y}_1(t) &= y_4(t) \\
\dot{y}_2(t) &= y_5(t) \\
\dot{y}_3(t) &= Aa\,\mathrm{Sigm}[y_1(t) - y_2(t)] - 2ay_3(t) - a^2 y_0(t) \\
\dot{y}_4(t) &= Aa\{I(t) + C_2\,\mathrm{Sigm}[C_1 y_0(t)]\} - 2ay_4(t) - a^2 y_1(t) \\
\dot{y}_5(t) &= BbC_4\,\mathrm{Sigm}[C_3 y_0(t)] - 2by_5(t) - b^2 y_2(t) \ ,
\end{aligned}
\tag{9}
$$

where $y_0$ accounts for the ePSP generated by the PNs, $y_1$ is the sum of the ePSP generated by the rPNs and the external excitatory inputs, and $y_2$ is the iPSP generated by the INs. Finally, the mean membrane potential of the pyramidal population is given by $v \coloneqq y_1 - y_2$, which we use as the main observable throughout this paper.

**Network model.**   We consider a network composed of $N$ nodes, each representing a brain region. The dynamics of each node follows the Jansen model for a cortical column. Therefore, the 6 equations governing the dynamics of node $i$ read [35]

$$
\begin{aligned}
\dot{y}_{0,i}(t) &= y_{3,i}(t) \\
\dot{y}_{1,i}(t) &= y_{4,i}(t) \\
\dot{y}_{2,i}(t) &= y_{5,i}(t) \\
\dot{y}_{3,i}(t) &= Aa\,\mathrm{Sigm}[y_{1,i}(t) - y_{2,i}(t)] - 2ay_{3,i}(t) - a^2 y_{0,i}(t) \\
\dot{y}_{4,i}(t) &= Aa\{I_i(t) + C_2\,\mathrm{Sigm}[C_1 y_{0,i}(t)]\} - 2ay_{4,i}(t) - a^2 y_{1,i}(t) \\
\dot{y}_{5,i}(t) &= BbC_4\,\mathrm{Sigm}[C_3 y_{0,i}(t)] - 2by_{5,i}(t) - b^2 y_{2,i}(t) \ .
\end{aligned}
\tag{10}
$$

The quantity $I_i$ accounts for the incoming signals from the rest of the network or other layers not represented in the model. In the brain model we consider that the different regions are coupled only through excitation, thus inhibition acts only locally. Also, all regions receive an external input from subcortical regions not represented in our model, in the form of a

common constant firing rate $p$. Altogether, $I_i(t)$ takes the form of the sum of two independent terms:

$$I_i(t) = p + \epsilon \sum_{j=1}^{N} \tilde{W}_{ij} \, \text{Sigm}[y_{1,j}(t) - y_{2,j}(t)] \, , \tag{11}$$

where $\epsilon$ is the coupling strength, and $\tilde{W} = (\tilde{w}_{ij})$ is the row-normalized structural connectivity matrix (see next section). We study the system under the influence of the external driving firing rate $p$ and the coupling strength $\epsilon$, leaving all the other parameters fixed, as defined in Table 1. As mentioned above, from the six variables that characterize the dynamics of each brain region, we monitor the mean membrane potential of the pyramidal neurons, $v_i := y_{1,i} - y_{2,i}$.

**Normalized connectivity.** Given a $N \times N$ structural connectivity (SC) matrix $W$, a detailed mathematical analysis of system (10) is generally unfeasible. In order to allow for a semi-analytical treatment, we consider a normalized version of the topology. Such normalization is usually employed in large-scale brain models [30, 33, 35, 44, 45, 56, 57]. Let $D = (d_{ij})$ be a diagonal $N \times N$ matrix whose non-zero entries are the sum of incoming connections to each node:

$$d_{ij} = \begin{cases} s_i & i = j \\ 0 & i \neq j \end{cases} \tag{12}$$

where

$$s_i = \sum_{j=1}^{N} w_{ij} \tag{13}$$

is the in-strength of node $i$. Then we consider the *row-normalized* SC matrix

$$\tilde{W} = (\tilde{w}_{ij}) := D^{-1} W \tag{14}$$

whose elements are $\tilde{w}_{ij} := w_{ij}/s_i$, i.e., $\tilde{W}$ is obtained dividing each row of $W$ by its sum. Therefore, the sum of the elements on each row equals unity i.e.,

$$\sum_{j=1}^{N} \tilde{w}_{ij} = 1 \, . \tag{15}$$

**Table 1. Parameters of the Jansen system.**

| Parameter | Meaning | Value |
|:---:|:---:|:---:|
| $A$ | Maximal amplitude of excitatory post-synaptic potentials | 3.25mV |
| $B$ | Maximal amplitude of inhibitory post-synaptic potentials | 22mV |
| $a$ | Characteristic decay time for ePSP | $100\text{s}^{-1}$ |
| $b$ | Characteristic decay time for iPSP | $50\text{s}^{-1}$ |
| $C_1, C_2, C_3, C_4$ | Synaptic strength (average number of synapses) between populations | 135, 108, 33.75, 33.75 |
| $e_0$ | Half of the maximum firing rate | 2.5Hz |
| $v_0$ | Potential where half the maximum firing rate is achieved | 6mV |
| $r$ | Neuronal excitability | $0.56\text{mV}^{-1}$ |
| $\tilde{w}_{ij}$ | Connectivity weights | from data |
| $p$ | Constant baseline firing rate to pyramidal neurons | not fixed (Hz) |
| $\epsilon$ | Coupling strength | not fixed |

Matrices with this normalization are sometimes called *right stochastic matrices* [111]. An important property of right stochastic matrices that we use in our analysis is that their largest eigenvalue is exactly $\Lambda_1 = 1$, which corresponds to a uniform eigenvector $\phi^{(1)} := (1, \ldots, 1)^T$. By the by the Gershgorin circle theorem [112], all other eigenvalues are bounded within the unit circle.

Finally, we remark that since the original structural connectome $W$, is symmetric, so is the matrix $Z := D^{-\frac{1}{2}}WD^{-\frac{1}{2}}$. Hence $Z$ is diagonalizable and has real eigenvalues. Using that $W = D\tilde{W}$ we obtain

$$Z := D^{\frac{1}{2}}\tilde{W}D^{-\frac{1}{2}}. \tag{16}$$

Therefore, $Z$ and $\tilde{W}$ are similar (in the mathematical sense), i.e., they share the same eigenvalues. The fact that the eigenvalues o $\tilde{W}$ are real (due to the symmetry of $W$) is not strictly necessary to carry our analysis based on the MSF, but it simplifies it.

## Homogeneous states

A row-normalized connectivity matrix ensures that all the different units in the network receive the same amount of input, although distributed differently across the different nodes. Using such type of connectivities, it is always possible to find *homogeneous* or *uniform* solutions -either stationary or time dependent- in which all nodes of the network behave identically.

Let $y_0, \ldots, y_5$ be the variables that characterize the dynamics of each brain region in a homogeneous state. Imposing then $y_{m,i}(t) = y_m(t)$ for $m = 0, \ldots, 5$ and $i = 1, \ldots, N$ one finds that the incoming input for each brain region in (10) reads

$$I_i(t) = p + \epsilon\sum_{j=1}^{N}\tilde{w}_{ij}\,\mathrm{Sigm}[y_{1,j}(t) - y_{2,j}(t)] = p + \epsilon\,\mathrm{Sigm}[y_1(t) - y_2(t)]\,, \tag{17}$$

thus it does not depend on the node index $i$ anymore. Replacing this expression of the input in the Jansen model (10) one finds that, in a homogeneous state, the equations for the evolution of each network node read

$$\dot{y}_0(t) = y_3(t)$$

$$\dot{y}_1(t) = y_4(t)$$

$$\dot{y}_2(t) = y_5(t)$$

$$\dot{y}_3(t) = Aa\,\mathrm{Sigm}[y_1(t) - y_2(t)] - 2ay_3(t) - a^2y_0(t) \tag{18}$$

$$\dot{y}_4(t) = Aa\{p + \epsilon\,\mathrm{Sigm}[y_1(t) - y_2(t)] + C_2\,\mathrm{Sigm}[C_1y_0(t)]\} - 2ay_4(t) - a^2y_1(t)$$

$$\dot{y}_5(t) = BbC_4\,\mathrm{Sigm}[C_3y_0(t)] - 2by_5(t) - b^2y_2(t)\,.$$

Therefore, this low-dimensional system determines all homogeneous states of the coupled system (10). Moreover, this system also retains the stability of such homogeneous states subject to uniform perturbations, i.e., perturbations that act identically at each brain region, and therefore do not change the homogeneous character of the trajectories. In other words, Eq (18) define an invariant manifold of Eq (10). Nonetheless, stable states in the homogeneous invariant manifold might still be unstable to heterogeneous perturbations, i.e., perturbations transverse to the manifold.

## Transverse stability

The following stability analysis is common in the study of dynamical systems in complex networks, specially (but not only), in the context of diffusive coupling [54, 68, 69]. The same technique can be applied to both, homogeneous fixed points and homogeneous limit-cycles. In the later case it is known as the Master Stability Function (MSF) [58] (see also [59, 61] for introductory reviews). In this section we explain such stability analysis on the Jansen model, but it can be easily extended to other systems. We use bold symbols for $N$ and $6N$ dimensional vectors and matrices, and regular symbols for 6-dimensional vectors, 6-dimensional matrices, and scalar quantities. All scalar quantities except the time $t$ and the coupling strength $\epsilon$ have a subscript.

Let $y^{(0)} := (y_0^{(0)}, \ldots, y_5^{(0)})^T$ be a solution of (18). Then, the $6N$-dimensional vector

$$\boldsymbol{y}^{(0)} = (y_0^{(0)}, \ldots, y_0^{(0)}, \overbrace{\ldots}^{N}, y_0^{(0)}, \ldots, y_5^{(0)})^T \tag{19}$$

is a homogeneous solution of (10), either stationary or periodic. Let us consider an arbitrary small perturbation

$$\boldsymbol{\delta y} = (\delta y_{0,1}, \ldots, \delta y_{5,1}, \ldots, \delta y_{0,N}, \ldots, \delta y_{5,N})^T \tag{20}$$

where each component $\delta y_{k,j}$ is the perturbation acting on the variable $k$ of node $j$ in the system (10). Expanding the velocity fields of (10) up to first order around $\boldsymbol{y}^{(0)}$ one obtains the linear evolution of the perturbation vector as

$$\frac{d}{dt}\boldsymbol{\delta y}(t) = \boldsymbol{J}(\boldsymbol{y}^{(0)})\boldsymbol{\delta y}(t) \tag{21}$$

where $\boldsymbol{J}$ is the full $6N \times 6N$ Jacobian of system (10), evaluated at $\boldsymbol{y}^{(0)}$. Notice that $\boldsymbol{J}$ can be written as

$$\boldsymbol{J}(\boldsymbol{y}^{(0)}) = J(y^{(0)}) \otimes \boldsymbol{I}_N + \epsilon K(y^{(0)}) \otimes \tilde{\boldsymbol{W}} \tag{22}$$

where $\otimes$ denotes the Kronecker product, $J$ is the $6 \times 6$ Jacobian of the uncoupled Jansen Eq (9), $\boldsymbol{I}_N$ is the $N \times N$ identity matrix, and $K = (k_{ij})$ is a $6 \times 6$ matrix defined by

$$k_{ij} = \begin{cases} Aa\,\mathrm{Sigm}'(y_1 - y_2) & i = 5, j = 2 \\ -Aa\,\mathrm{Sigm}'(y_1 - y_2) & i = 5, j = 3 \\ 0 & \text{otherwise .} \end{cases} \tag{23}$$

One could, in principle, evaluate numerically the eigenvalues and eigenvectors of this Jacobian in order to obtain the stability properties of the system. Nonetheless, there is a simpler and more informative approach based on expressing the perturbation vector $\boldsymbol{\delta y}$ in an adequate coordinate system.

Let $\boldsymbol{\Phi}^{(\alpha)} = (\phi_1^{(\alpha)}, \ldots, \phi_N^{(\alpha)})^T$ be a normalized eigenvector of $\tilde{\boldsymbol{W}}$ associated with the eigenvalue $\Lambda_\alpha$ for $\alpha = 1, \ldots, N$, so that

$$\tilde{\boldsymbol{W}}\boldsymbol{\Phi}^{(\alpha)} = \Lambda_\alpha\boldsymbol{\Phi}^{(\alpha)} . \tag{24}$$

The set of eigenvectors $\{\boldsymbol{\Phi}^{(1)}, \ldots, \boldsymbol{\Phi}^{(N)}\}$ constitute a basis of the vector space $\mathbb{R}^N$, thus we can express the perturbation $\boldsymbol{\delta y}$ of the homogeneous solution as a linear combination of such

vectors. In this new basis, the perturbation acting on variable $k$ of the $j$th node reads

$$\delta y_{k,j}(t) = \sum_{\alpha=1}^{N} u_k^{(\alpha)}(t)\phi_j^{(\alpha)} \tag{25}$$

where $u^{(\alpha)}(t) = (u_0^{(\alpha)}(t), \ldots, u_5^{(\alpha)}(t))^T$ are the coordinates of the perturbation vector expressed in the new basis. This can also be expressed in vectorial form using the Kronecker operator $\otimes$ as

$$\boldsymbol{\delta y}(t) = \sum_{\alpha=1}^{N} u^{(\alpha)}(t) \otimes \boldsymbol{\Phi}^{(\alpha)} . \tag{26}$$

Now it is necessary to perform some calculations using the tensor product $\otimes$. For simplicity we drop some dependences: $J = J(y^{(0)})$, $K = K(y^{(0)})$, and $u^{(\alpha)} = u^{(\alpha)}(t)$. Applying the change of coordinates on Eq (21) and using Eq (22), we have that

$$
\begin{aligned}
\frac{d}{dt}\boldsymbol{\delta y} \quad &= \frac{d}{dt}\sum_{\alpha=1}^{N} u^{(\alpha)} \otimes \boldsymbol{\Phi}^{(\alpha)} \\[2mm]
&= \sum_{\alpha=1}^{N} \left(\frac{d}{dt}u^{(\alpha)}\right) \otimes \boldsymbol{\Phi}^{(\alpha)} \\[2mm]
&= (J \otimes \boldsymbol{I}_N + \epsilon K \otimes \tilde{\boldsymbol{W}})\sum_{\alpha=1}^{N} u^{(\alpha)} \otimes \boldsymbol{\Phi}^{(\alpha)} \\[2mm]
&= (J \otimes \boldsymbol{I}_N)\left(\sum_{\alpha=1}^{N} u^{(\alpha)} \otimes \boldsymbol{\Phi}^{(\alpha)}\right) + \epsilon(K \otimes \tilde{\boldsymbol{W}})\left(\sum_{\alpha=1}^{N} u^{(\alpha)} \otimes \boldsymbol{\Phi}^{(\alpha)}\right) \\[2mm]
&= \sum_{\alpha=1}^{N}(J \otimes \boldsymbol{I}_N)(u^{(\alpha)} \otimes \boldsymbol{\Phi}^{(\alpha)}) + \epsilon\sum_{\alpha=1}^{N}(K \otimes \tilde{\boldsymbol{W}})(u^{(\alpha)} \otimes \boldsymbol{\Phi}^{(\alpha)}) \\[2mm]
&= \sum_{\alpha=1}^{N}(Ju^{(\alpha)}) \otimes (\boldsymbol{I}_N\boldsymbol{\Phi}^{(\alpha)}) + \epsilon\sum_{\alpha=1}^{N}(Ku^{(\alpha)}) \otimes (\tilde{\boldsymbol{W}}\Phi^{(\alpha)}) \\[2mm]
&= \sum_{\alpha=1}^{N}(Ju^{(\alpha)}) \otimes \boldsymbol{\Phi}^{(\alpha)} + \epsilon\sum_{\alpha=1}^{N}(Ku^{(\alpha)}) \otimes (\Lambda_\alpha\boldsymbol{\Phi}^{(\alpha)}) \\[2mm]
&= \sum_{\alpha=1}^{N}(Ju^{(\alpha)}) \otimes \boldsymbol{\Phi}^{(\alpha)} + \epsilon\sum_{\alpha=1}^{N}(\Lambda_\alpha Ku^{(\alpha)}) \otimes \boldsymbol{\Phi}^{(\alpha)} \\[2mm]
&= \sum_{\alpha=1}^{N}(J + \epsilon\Lambda_\alpha K)u^{(\alpha)} \otimes \boldsymbol{\Phi}^{(\alpha)} ,
\end{aligned}
\tag{27}
$$

where we have used the diagonalization of the connectivity matrix Eq (24). Now, making use of the linear independence of the eigenvectors $\{\boldsymbol{\Phi}^{(\alpha)}\}_{\alpha=1}^{N}$ one obtains that the evolution of $u^{(\alpha)}(t)$ becomes independent for each $\alpha = 1, \ldots, N$ through the relation

$$\dot{u}^{(\alpha)} = (J + \epsilon\Lambda_\alpha K)u^{(\alpha)} , \tag{28}$$

where $\mathcal{J}(y^{(0)}; \Lambda) := J + \epsilon\Lambda_\alpha K$ is a family of $6 \times 6$ Jacobians that depend on the homogeneous state of the system $y^{(0)}$ and on the structural connectivity eigenvalues $\Lambda_\alpha$. If $y^{(0)}$ is a fixed point,

the stability of the homogeneous solution simplifies to studying the eigenvalues (and eigenvectors) of the Jacobian $\mathcal{J}$, which is a function of the connectivity matrix eigenvalues $\Lambda_\alpha$. If $y^{(0)} = y^{(0)}(t)$ is a periodic solution, then $\mathcal{J}$ is a periodic matrix and Floquet theory applies [72]. In this context, the growth rate of a perturbation acting at the limit-cycle solution is determined by the real part of the *Floquet exponents* corresponding to $\mathcal{J}$, which must be determined numerically. Practically speaking, for each value of $\Lambda_\alpha$, we compute the real part of the largest Floquet exponent of $\mathcal{J}$ as the largest Lyapunov exponent of the linear system Eq (28). The code is available in github (www.github.com/pclus/transverse-instabilities). Finally, in order to analyze the contribution of each structural eigenmode $\alpha$ on the growth rate of a specific perturbation vector $\boldsymbol{\delta y}$ we compute the quantity

$$c_\alpha = \mu_\alpha \|u^{(\alpha)}\| . \tag{29}$$

## Lyapunov exponents

Lyapunov exponents (LE) provide the growth rate of small perturbations acting on a time-evolving trajectory of a dynamical system [77, 113]. To compute these quantities in practice we employ a dynamical algorithm based on QR-decompositions, using Householder reflections [77]. The algorithm is embedded in the available software (www.github.com/pclus/transverse-instabilities). There are as many Lyapunov exponents as system dimensions, and they are usually sorted from largest to smallest: $\lambda_1 \geq \lambda_2 \geq \lambda_3 \geq \ldots \geq \lambda_{6N}$. Using the two largest Lyapunov exponents $\lambda_1$ and $\lambda_2$, we can classify the system state in 5 types:

- Fixed points, corresponding to both LE being negative ($\lambda_1, \lambda_2 < 0$).

- Periodic dynamics, identified by a zero LE and the rest being negative ($\lambda_1 = 0, \lambda_2 < 0$).

- Quasiperiodic dynamics, i.e., a regime in which the system evolves with at least two incommensurate characteristic frequencies. In this case both LE are zero ($\lambda_1 = 0, \lambda_2 = 0$).

- Chaotic dynamics, with a single positive LE ($\lambda_1 > 0, \lambda_2 \leq 0$).

- Hyperchaos, with more than one positive LE ($\lambda_1, \lambda_2 > 0$).

Since the Lyapunov exponents are computed numerically, we need to impose a threshold to discern between zero and non-zero values. We found that a value of $|\lambda_k| < 10^{-4}$ was a reasonable cut-off.

Another useful application of Lyapunov exponents is to compute the dimensionality of the attractor, which is fractal in chaotic states. The Kaplan-Yorke formula [80] provides an approximation of the fractal dimension of an attractor as

$$\mathcal{D}_{\mathrm{KY}} = j + \frac{\sum_{i=1}^{j} \lambda_i}{|\lambda_{j+1}|} , \tag{30}$$

where $j$ is the LE for which

$$\sum_{i=1}^{j} \lambda_i \geq 0 \quad \text{and} \quad \sum_{i=1}^{j+1} \lambda_i < 0 . \tag{31}$$

## Synchronization and wave-propagation analysis

We quantify the synchronization of the network by extracting the phase variables $\phi_j(t)$ from each node's dynamics $v_j(t)$ using the Hilbert transform [114], i.e.,

$$\phi_j(t) = \arctan\left(\frac{v_j^{(H)}(t)}{v_j(t)}\right)$$

where $v_j^{(H)}(t)$ is the Hilbert transform of $v_j(t)$,

$$v_j^{(H)}(t) = \frac{1}{\pi}\mathrm{P.V.}\int_{-\infty}^{\infty}\frac{v_j(t)}{t-\tau}d\tau$$

with P.V. indicating that the integral corresponds to the Cauchy principal value.

Then we can compute the complex quantity

$$Z(t) = R(t)e^{i\Psi(t)} := \sum_{j=1}^{N}e^{i\phi_j(t)} \tag{32}$$

in which $R \in [0, 1]$ is the usual Kuramoto order parameter, and $\Psi$ is the collective phase variable [53, 114].

From a given phase gradient in space, we extract the propagation vector of each node $\zeta_j$ by following the same procedure as in [30] (see also [5]). Briefly, the method consists on numerically finding the temporal and spatial derivatives of $\phi$ and then solving

$$\frac{d}{dt}\phi = \boldsymbol{\nabla}\phi \cdot \zeta + \frac{\partial\phi}{\partial t} = 0 \tag{33}$$

for $\zeta$. The numerical differentiation of the phase is obtained through a constrained natural element method [73]. The *instantaneous speed* of each node is then given by the modulus of the velocity vector, $\zeta := \|\zeta\|$.

In order to measure the degree of alignment of a certain direction vector with neighbouring nodes, we use a measure of *local polarization*. Let $\mathcal{N}_j$ be the set of nodes that are located at less than 40mm than node $j$. Then the local polarization of $j$ is defined as

$$a_j = \frac{1}{|\mathcal{N}_j|}\left\|\sum_{k\in\mathcal{N}_j}\frac{\zeta_k}{\zeta_k}\right\| \in [0, 1] . \tag{34}$$

A value close to 1 indicates a perfect alignment of node $j$ with its neighbours, whereas a value close to zero indicates a nearly isotropic propagation at $j$.

Finally, in order to assess how much the direction of propagation changes over time we compute the *directional coherence* as the modulus of the time-average normalized vectors

$$\rho_j = \left\|\left\langle\frac{\zeta_j}{\zeta_j}\right\rangle\right\| . \tag{35}$$

Notice that $\rho_j$ corresponds to the mean-resultant length of the temporal distribution of directions $\zeta_j/\zeta_j$.

## Next generation neural mass model

The firing rate equations for populations of quadratic integrate-and-fire neurons are based on an exact mean-field theory derived by Montbrió et. al. [17]. This theory has been recently

shown to apply also for neurons subject to Cauchy white noise [115]. Here we consider a PING mechanism in which the oscillatory activity of each brain region is in the gamma range, similarly to the scenario studied in [83]. Hence, at each network node $j$ we consider an excitatory and a inhibitory populations of QIF neurons, each characterized by its firing rate, $r$, and mean membrane potential, $v$. We also consider delta pulses for the PSP, as a means to illustrate that the shape of the PSP is not crucial for the transverse instability of the synchronized state. Hence, the neural activity of brain region $j$ is given by

$$\tau \dot{r}_{e,j} = \frac{\Delta}{\tau \pi} + 2r_{e,j}v_{e,j}$$

$$\tau \dot{v}_{e,j} = \eta_e - (\pi r_{e,j}\tau)^2 + v_{e,j}^2 + \tau(J_{ee}r_{e,j} - J_{ie}r_{i,j} + I_j)$$

$$\tau \dot{r}_{i,j} = \frac{\Delta}{\tau \pi} + 2r_{i,j}v_{i,j}$$

$$\tau \dot{v}_{i,j} = \eta_i - (\pi r_{i,j}\tau)^2 + v_{i,j}^2 + \tau(J_{ei}r_{e,j} - J_{ii}r_{i,j})$$

(36)

where $r_{e,j}$ and $v_{e,j}$ (resp. $r_{e,j}$, $v_{e,j}$) are the firing rate and mean-membrane potential of the excitatory (resp. inhibitory) population of region $j$. The explanation and value of the different system parameters are given in Table 2. This setup corresponds to the PING mechanism for gamma activity, and shows that a single uncoupled brain area displays activity at a frequency around 40Hz [83]. We have verified that the results are robust upon considering other parameter choices and, in particular, when including recurrent inhibition ($J_{ii} > 0$). As in the network of Jansen NMMs, we consider that different regions interact only through excitation, i.e.,

$$I_j = \epsilon \sum_{j=1}^{N} \tilde{w}_{jk} r_{e,k} \; .$$

(37)

Following the same argument as for the Jansen system, if all nodes of the network evolve identically, then the dynamics of each network node follows

$$\tau \dot{r}_e = \frac{\Delta}{\tau \pi} + 2r_e v_e$$

$$\tau \dot{v}_e = \eta_e - (\pi r_e \tau)^2 + v_e^2 + \tau((J_{ee} + \epsilon)r_e - J_{ie}r_i)$$

$$\tau \dot{r}_i = \frac{\Delta}{\tau \pi} + 2r_i v_i$$

$$\tau \dot{v}_i = \eta_i - (\pi r_i \tau)^2 + v_i^2 + \tau(J_{ei}r_e - J_{ii}r_i) \; .$$

(38)

**Table 2. Parameters of the NG-NMM.**

| Parameter | Meaning | Value |
|---|---|---|
| $\tau$ | Time constant | 20ms |
| $\eta_e$ | Baseline constant current for excitatory neurons | not fixed |
| $\eta_i$ | Baseline constant current for inhibitory neurons | -5 |
| $\Delta$ | Single neuron noise intensity | 1 |
| $J_{ee}, J_{ei}, J_{ii}, J_{ie}$ | Synaptic strength between populations | 5,12,0,18 |
| $\epsilon$ | Coupling strength | not fixed |

### Numerical simulations and root-finding algorithm

Numerical simulations of the systems have been performed with a fourth-order Runge-Kutta algorithm with time step $\Delta t = 10^{-3}$. The C code to simulate the model composed of Jansen NMMs is available at www.github.com/pclus/transverse-instabilities. Most of the simulations have been run for 1000s after discarding an extra 1000s of relaxation time. The only exception is Fig 5(a), for which we used a relaxation time of $10^4$s in order to achieve a good convergence of the zero Lyapunov-exponents needed to compute the Kaplan-Yorke dimension.

Two different types of initial conditions have been considered:

- **Close to homogeneous**: we select a point in the trajectory of the homogeneous dynamics of the system using the equations of the homogeneous manifold, Eqs (18) and (38), and adding an independent random perturbation to each variable, uniformly distributed between $-10^{-3}$ to $10^{-3}$.

- **Random**: all variables of the system randomly distributed between $-1$ and $1$.

In order to obtain the stationary states of the unnormalized network topology $W$ in Section entitled "Normalized connectivity" above, we use the adaptive Newton-Raphson multidimensional root-finding algorithm provided by the GNU Scientific Library [116].

## Supporting information

**S1 Movie. Travelling wave in the large-scale brain model.** Spatial representation of the mean membrane potential from simulations of the model corresponding to the results of Fig 3(a). (MP4)

**S2 Movie. Travelling wave in the large-scale brain model.** Spatial representation of the mean membrane potential from simulations of the model corresponding to the results of Fig 3(b). (MP4)

**S3 Movie. Chaotic dynamics the large-scale brain model.** Spatial representation of the mean membrane potential from simulations of the model corresponding to the results of Fig 4. (MP4)

**S4 Movie. Periodic dynamics the large-scale brain model without normalization.** Spatial representation of the mean membrane potential from simulations of the model corresponding to the results of Fig 6 ($p = 320$ and $\epsilon = 50$). (MP4)

**S5 Movie. Chaotic dynamics the large-scale brain model without normalization.** Spatial representation of the mean membrane potential from simulations of the model corresponding to the results of Fig 6 ($p = 170$ and $\epsilon = 50$). (MP4)

## Author Contributions

**Conceptualization:** Pau Clusella, Gustavo Deco, Giulio Ruffini, Jordi Garcia-Ojalvo.

**Data curation:** Gustavo Deco, Morten L. Kringelbach.

**Formal analysis:** Pau Clusella, Jordi Garcia-Ojalvo.

**Funding acquisition:** Gustavo Deco, Giulio Ruffini, Jordi Garcia-Ojalvo.

**Investigation:** Pau Clusella, Gustavo Deco, Giulio Ruffini, Jordi Garcia-Ojalvo.

**Methodology:** Pau Clusella.

**Resources:** Morten L. Kringelbach.

**Software:** Pau Clusella.

**Supervision:** Giulio Ruffini, Jordi Garcia-Ojalvo.

**Validation:** Pau Clusella.

**Visualization:** Pau Clusella.

**Writing – original draft:** Pau Clusella, Jordi Garcia-Ojalvo.

**Writing – review & editing:** Pau Clusella, Gustavo Deco, Morten L. Kringelbach, Giulio Ruffini, Jordi Garcia-Ojalvo.

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
