## [Decision Letter · Decision Letter 0]

5 Feb 2023

Dear Dr Clusella,

Thank you very much for submitting your manuscript "Complex spatiotemporal oscillations emerge from transverse instabilities in large-scale brain networks" for consideration at PLOS Computational Biology.

As with all papers reviewed by the journal, your manuscript was reviewed by members of the editorial board and by several independent reviewers. In light of the reviews (below this email), we would like to invite the resubmission of a significantly-revised version that takes into account the reviewers' comments.

We cannot make any decision about publication until we have seen the revised manuscript and your response to the reviewers' comments. Your revised manuscript is also likely to be sent to reviewers for further evaluation.

Sincerely,

Boris S. Gutkin

Academic Editor

PLOS Computational Biology

Marieke van Vugt

Section Editor

PLOS Computational Biology

Reviewer's Responses to Questions

**Comments to the Authors:**

Reviewer #1: The work by Clusella et al. studies the dynamics of large-scale brain networks. These networks are build using the topology reported in Ref. [60] (Automated Anatomical parcellation into 90 brain areas). The local dynamics is incorporated through the Jansen(-Rit) model, and additional simulations with a next-generation neural mass model are also carried out in order to assess the generality of the results.

As the title of the paper emphasizes, transverse instabilities of the homogeneous state are informative concerning the complex spatiotemporal oscillations of the models. The semi-analytical analysis uses the master stability function method. Complementary, Lyapunov analysis is used to characterize complex states.

Results are, to a certain degree, independent of the details, such as the specific neural mass model type, or the zero-raw constraint of the coupling matrix.

To the best of my knowledge, the results are new and interesting. The analysis is accurate and extensive. The text is well written, and the figures are informative. I recommend the publication of this manuscript, once the authors react to my minor comments:

1) Page 6, 15, banish -> vanish.

2) Lyapunov exponents have dimension of inverse of time. Units should appear in Fig. 5(a), and in page 11 in the value of 1/lambda_1.

3) Page 11, lines 34, I'd write m/s instead of ms^{-1}, to avoid confusion with the inverse of a millisecond.

4) Page 11. The formula for the mean resultant length appears to be incomplete.

5) Page 25, I couldn't decipher equation (29).

6) Page 26, I'd write down the formula of the Hilbert transform.

7) Equation (24) is inconsistent, as it equates a scalar to a vector.

8) Same criticism for Eq. (35).

9) Eq (37), epsilon is already included in Eq. (36).

9) Equation (38). I_j stands for r_e.

Reviewer #2: The present manuscript investigates the stability of synchronized oscillations on a large scale brain connectivity network. The dynamics of each node of the network is described by a neural mass model (NMM), the Jansen-Rit model in most of the manuscript with two excitatory populations and one inhibitory population or the Montbrio et al’s mean field model in the last part with one excitatory population and one inhibitory population. The coupling between nodes is through excitatory coupling among the excitatory populations suitably normalized so that a fully synchronized state exists. The authors find that this fully synchronized state is not stable in parts of the parameter space and characterize the more complex spatio-temporal dynamics that appear then. Interestingly, they also find that different types of travelling waves can be supported on the network for given parameter values, namely, multistability of traveling waves. The manuscript is clearly written, the computations appear competently performed and present interesting results.

I have however several concerns that I would like the authors to address:

1) The synchronization of oscillations in NMMs has been investigated on simpler networks in previous works that I believe are quite relevant for the finding of the manuscript and should be cited.

In early work, Ermentrout, G. B., and N. Kopell. "Multiple pulse interactions and averaging in systems of coupled neural oscillators." Journal of Mathematical Biology 29.3 (1991): 195-217 showed that excitatory coupling can lead to instability of in-phase synchronization. The variety of dynamical behaviors that ensue in a two node excitory-inhibitory networks was investigated in Borisyuk, G. N., et al. "Dynamics and bifurcations of two coupled neural oscillators with different connection types." Bulletin of mathematical biology 57 (1995): 809-840 including many periodic regime and a scenario for chaos. More recently, the stability of synchronized oscillations in a chain of oscillatory excitatory-inhibitory networks was investigated by Floquet techniques (the MSF of the present manuscript) in Kulkarni, A. et al "Synchronization, stochasticity, and phase waves in neuronal networks with spatially-structured connectivity." Frontiers in computational neuroscience 14 (2020): 569644. The instability of the two-node network at small coupling was reported to translate into a long-wave instability similarly to the “transverse instability” reported in Fig.2b of the present manuscript.

2) Importantly, in these previous works, it was found that the tranverse instability was quite sensitive to the type of synaptic connections between different network nodes. Even if one supposes, that different nodes connect by excitatory connections, it is a very strong assumption that the incoming excitatory connections only target the excitatory populations. In Kulkarni et al, it is in fact shown that the transverse instability disappears when the excitatory and inhibitory populations on a node are targeted in a similar fashion by the local and long-range excitatory connections. The authors should either convincingly argue why it is legitimate to neglect the long-range excitatory connections on interneurons or, better, investigate how their results are modified of they do not make this assumption.

3) Other issues that the authors should address, or at least discuss, is the biological realism of some of their other choices.

a) What is the impact of their neglect the local recurrent interactions between interneurons?

b) An estimate of the coupling strength epsilon that one expects for the brain would also be welcome.

c) If I understand well, the authors starts from a brain network with symmetric connections before row normalization. Moreover, they assume an identical NMM on each node. Other groups, and prominently X-J Wang’s have argued otherwise. They found more appropriate to use Kennedy’s data and moreover reported results consistent with a hierarchy of time scales in different areas as recently reviewed in Wang, X.-J. "Theory of the multiregional neocortex: large-scale neural dynamics and distributed cognition." Annual review of neuroscience 45 (2022): 533-560. These issues should be at least discussed.

d) Finally, it would be most welcome that the authors compare their results in a more detailed fashion to the available recordings. Are there features in their simulations that agree with what is found in the data (for instance, similar dephasing of oscillations in the model and the recordings). In this respect, alpha-waves in the brain usually do not appear as sustained oscillations but as transient oscillatory bursts. This is presumably not what is seen in the present model. This may suggest that the brain is actually functioning in a stable but fluctuating regime close to an oscillatory boundary. The authors should also discuss this issue.

Minor concerns:

- The authors should define the abbreviations they use (e.g.EEG, fMRI, NMM).

**Have the authors made all data and (if applicable) computational code underlying the findings in their manuscript fully available?**

Reviewer #1: Yes

Reviewer #2: Yes

PLOS authors have the option to publish the peer review history of their article (what does this mean?). If published, this will include your full peer review and any attached files.

Reviewer #1: No

Reviewer #2: No
---

## [Decision Letter · Decision Letter 1]

24 Mar 2023

Dear Dr Clusella,

We are pleased to inform you that your manuscript 'Complex spatiotemporal oscillations emerge from transverse instabilities in large-scale brain networks' has been provisionally accepted for publication in PLOS Computational Biology.

Best regards,

Boris S. Gutkin

Academic Editor

PLOS Computational Biology

Marieke van Vugt

Section Editor

PLOS Computational Biology

Reviewer's Responses to Questions

**Comments to the Authors:**

Reviewer #1: I recommend publication of the manuscript in its present form.

Reviewer #2: The authors have taken into account my comments. They have performed additional work that answers some of my questions. i believe that the manuscript is now suitable for publication in PCBI.

**Have the authors made all data and (if applicable) computational code underlying the findings in their manuscript fully available?**

Reviewer #1: Yes

Reviewer #2: Yes

PLOS authors have the option to publish the peer review history of their article (what does this mean?). If published, this will include your full peer review and any attached files.

Reviewer #1: No

Reviewer #2: No

---

## [Editor Report · Acceptance letter]

5 Apr 2023

PCOMPBIOL-D-22-01758R1 

Complex spatiotemporal oscillations emerge from transverse instabilities in large-scale brain networks

Dear Dr Clusella,

I am pleased to inform you that your manuscript has been formally accepted for publication in PLOS Computational Biology. Your manuscript is now with our production department and you will be notified of the publication date in due course.

With kind regards,

Zsofi Zombor
